



# Towards accurate quantification of ice content in permafrost of the Central Andes, part I: geophysics-based estimates from three different regions

Christin Hilbich[1], Christian Hauck[1], Coline Mollaret[1], Pablo Wainstein[2], Lukas U. Arenson[3]

[1]Department of Geosciences, University of Fribourg, Fribourg, 1700, Switzerland
[2]BGC Engineering Inc., Calgary, AB, Canada
[3]BGC Engineering Inc., Vancouver, BC, Canada

*Correspondence to*: Christin Hilbich (christin.hilbich@unifr.ch)

**Abstract.** In view of the increasing water scarcity in the Central Andes in response to ongoing climate change, the significance of permafrost occurrences for the hydrological cycle is currently controversial. The lack of comprehensive field measurements and quantitative data on the local variability of internal structure and ground ice content further enhances the situation. We present field-based data from six extensive geophysical campaigns completed since 2016 in three different high-altitude regions of the Central Andes of Chile and Argentina (28 to 32° S). Our data cover various permafrost landforms ranging from ice-poor bedrock to ice-rich rock glaciers and are complemented by ground truthing information from boreholes and numerous test pits near the geophysical profiles. In addition to determining the thickness of the potential ice-rich layers from the individual profiles, we also use a quantitative 4-phase model to estimate the volumetric ground ice content in representative zones of the geophysical profiles.

The analysis of 52 geoelectrical and 24 refraction seismic profiles within this study confirmed that ice-rich permafrost is not restricted to rock glaciers, but is also observed in non-rock-glacier permafrost slopes in the form of interstitial ice as well as layers with excess ice, resulting in substantial ice contents. Consequently, non-rock glacier permafrost landforms, whose role for local hydrology has so far not been considered in remote-sensing based approaches, may be similarly relevant in terms of ground ice content on a catchment scale and should not be ignored when quantifying the potential hydrological significance of permafrost.

We state that geophysics-based estimates on ground ice content allow for more accurate assessments than purely remote-sensing-based approaches. The geophysical data can then be further used in upscaling studies to the catchment scale in order to reliably estimate the hydrological significance of permafrost within a catchment.



## 1    Introduction

Permafrost covers about 15 to 20 % of the Northern hemisphere global land-surface (Obu, 2021; Obu et al., 2019). Most
permafrost studies address permafrost occurrences either in the Arctic, Antarctica, or mountain ranges of the Northern
Hemisphere. However, in the mid-latitudes of the Southern Hemisphere the presence of permafrost is also widespread at high
elevation in the Central Andes of South America, where only few studies and even less borehole data (e.g., within the global
permafrost data base GTN-P, Biskaborn et al., 2019) currently exist. Continued climate change is projected to cause significant
temperature increase for the Subtropical Central Andes, yielding significant water shortage especially in the arid mountain
regions (Hock et al., 2019). In this context, the significance of permafrost occurrences in the Central Andes for the hydrological
cycle is currently controversial (e.g., Arenson and Jakob, 2010; Azócar and Brenning, 2010; Brenning, 2008; Duguay et al.,
2015; Jones et al., 2019). On the one hand, permafrost in general, and especially rock glaciers (which are conspicuous and
often ice-rich permafrost landforms) are considered key stores of frozen water and alternative future water resources in view
of the ongoing recession of glaciers in the dry Andes (e.g., García et al., 2017; Masiokas et al., 2020; Rangecroft et al., 2015).
Consequently, degrading permafrost is speculated to partly compensate the strongly decreasing glacial discharge in the future,
and aid the strong demand for fresh water from the Andean Cordillera for the growing population and economy in the Central
and dry Andes (Bradley et al., 2006; Schaffer et al., 2019).

On the other hand, the significance of permafrost for the hydrological cycle and regional hydrology is disputed (Arenson et
al., 2013; Duguay et al., 2015), due to (1) the methodology to quantify ground ice resources in Andean permafrost regions, (2)
the time scales involved for significant discharge from permafrost bodies, and also (3) unknowns on evaporation and
sublimation processes under intense solar radiation. A large part of the current debate focusses on rock glaciers as the most
prominent, ice-rich permafrost landforms, which can easily be identified by remote sensing (e.g., Azócar et al., 2017; Janke et
al., 2017; Rangecroft et al., 2014; Villarroel et al., 2018). Remote-sensing based rock glacier inventories have been used to
estimate the total ice volume in rock glaciers from rough estimates of rock glacier thickness (e.g., Rangecroft et al., 2015) or
empirical volume-area correlations (Brenning, 2005; Jones et al., 2018b, 2018a, 2019), with the aim to compare the ice content
stored in rock glaciers to the total ice content of glaciers per region. However, these estimates have been conducted without
any ground truthing or other means of validation, and volume-area correlations for rock glaciers have significant uncertainty
as local topography, geology and geomorphic processes are ignored. In addition, permafrost occurrences other than rock
glaciers have rarely been considered due to the difficulty of detecting them from space.

This focus on rock glaciers and remote-sensing-derived estimates points to the problem of a general lack of ground-based data
in the high Andes, with only very few exceptions (e.g., Arenson et al., 2010; Croce and Milana, 2002; Halla et al., 2021;
Monnier and Kinnard, 2013; de Pasquale et al., 2020). Numerous authors highlight the need for field observations regarding
thickness, internal structure, and ground ice content of permafrost, and especially rock glaciers, to evaluate the role of ground
ice within the hydrological cycle (Arenson et al., 2010; Azócar et al., 2017; Azócar and Brenning, 2010; Croce and Milana,
2002; Duguay et al., 2015; Jones et al., 2018a, 2019; Perucca and Angillieri, 2011; Rangecroft et al., 2015). Duguay et al.,



(2015) emphasise in this context that practically no quantitative data on the hydrology of rock glaciers are available, but that most studies are qualitative instead.

To estimate the ground ice content of permafrost landforms such as a rock glacier, both the total volume of the landform, i.e., horizontal and vertical extent, as well as the spatial variability of its ground ice content needs to be known. Both parameters can be derived from geophysical data. Compared to direct methods (core drillings or excavations/test pits), which are very costly and mostly restricted to point information or shallow depths, geophysical surveying can cover larger areas and depths, is cost-effective and comparatively easy to apply, and can be applied non-invasively, also in fragile and remote polar and high

mountain terrain (e.g., Kneisel et al., 2008). Recent developments in the application of geophysical techniques to permafrost problems have focused on quantitative estimates of volumetric ground ice content from electric, electromagnetic, seismic and gravimetric techniques, mostly applied in combination (Duvillard et al., 2018; Hauck et al., 2011; Hausmann et al., 2007; Mollaret et al., 2020; Oldenborger and LeBlanc, 2018; Wagner et al., 2019). Hauck et al. (2011), Wagner et al. (2019) and Mollaret et al. (2020) showed that the spatial distribution of the subsurface composition (ice, water, air and rock/soil content)

can be derived from linking the measured electrical and seismic properties through petrophysical models and validated their approach using borehole (core) data. In the Andes, such quantitative geophysical studies are still very rare and focused on individual rock glaciers (e.g., Halla et al., 2021; Monnier and Kinnard, 2013; de Pasquale et al., 2020).

To reduce the lack of comprehensive and quantitative field data on the local variability of ground ice content within rock glaciers but also on other ice-rich and ice-poor permafrost occurrences in the Central Andes, we conducted extensive

geophysical measurement campaigns in different high-altitude regions of Chile and Argentina. We here present a large number of geoelectric (Electrical Resistivity Tomography, ERT, 52 surveys) and seismic (Refraction Seismic Tomography, RST, 24 surveys) data sets from several permafrost sites with different geomorphologic settings, including numerous ice-rich and ice-poor permafrost occurrences (Table 1, Table 2). Borehole and test pit data are available for some of the sites, which are used to validate the quantitative estimates of ground ice contents by the 4-phase model (Hauck et al., 2011). The surveys were

conducted during the years 2016 - 2019 in four different regions between 28 and 32° S (Figure 1) in the framework of several Environmental Impact Assessment studies.

With these data, we want to address the following objectives: (1) Demonstrate the potential and feasibility of geophysical surveys for the quantification of ground ice content of different permafrost landforms in the Central Andes; (2) compare the ground ice content in different rock glaciers with non-rock glacier permafrost occurrences; and (3) analyse the uncertainties

of ground ice content estimates in the context of future studies of water availability from thawing permafrost under climate change.

In the following, we will introduce our methodology to estimate the thickness of ice-rich permafrost layers and quantify ground ice contents from geophysical surveys, present the compiled data set, and comment on the implications of the results regarding potential water storage within permafrost systems in high mountain regions.






**Table 1: Overview over main characteristics of the field locations and number and type of geophysical profiles (Abbreviations: RG = rock glacier, PR = protalus rampart, TS = talus slope, SED = sediment slopes (including gelifluction slopes, colluvial slopes, debris-covered bedrock, moraines, landslides).**

| Date | Location | Province | Elevation range [m] | nr. of profiles | | Landforms | | |
|------|----------|----------|---------------------|------|------|-------|------|------|
| | | | | ERT | RST | RG/PR | TS | SED |
| Feb 2016 | A | Choapa Province (CL) | 3500 - 3900 | 15 | 13 | 7 | 2 | 1 |
| Mar 2017 | B | Choapa Province (CL) | 3600- 3900 | 3 | 3 | 3 | - | - |
| Mar 2017 | C | Elqui Province (CL) | 4900 - 5100 | 8 | 2 | - | - | 8 |
| Feb 2018 | D | Copiapó Province (CL) San Juan Province (AR) | 5000 - 5200 | 10 | 3 | - | - | 10 |
| Feb 2019 | E | San Juan Province (AR) | 4200 - 4800 | 15 | 2 | 2 | 1 | 8 |
| Feb 2018 | F | San Juan Province (AR) | 4300 - 4500 | 1 | 1 | 1 | - | - |


**Figure 1: Map of the Central Andes with the study sites A - F, and detailed images for each of the study sites, showing the geophysical lines completed (Map data: © Google Earth 2021).**



## 2    Methods

With respect to the conducted ERT (Electrical Resistivity Tomography) and RST (Refraction Seismic Tomography) surveys,
we follow the well-established methodology described in Halla et al. (2021), Mewes et al. (2017), and Mollaret et al. (2019).
This methodology includes the conduction of the surveys, data processing with filtering of measured apparent resistivities
(ERT), first break picking (RST), data inversion using the software RES2DINV (Aarhus Geosoftware) and Reflex-W
(Sandmeier Geophysical Research) and, where applicable, running the 4-phase model (Hauck et al., 2011).

ERT data were obtained in the field using a SYSCAL multi-electrode instrument (Iris Instruments) with 48 electrodes. As ERT
data acquisition quality often suffers from low signal-to-noise ratios, induced by the high contact resistances of galvanically
coupled electrodes in dry and coarse-blocky substrates, all measurements were performed in the Wenner configuration to
ensure maximum signal strength. The spacing between the electrodes for the individual profiles varied between 1 and 8 m
depending on the desired survey geometry and penetration depth. The obtained apparent resistivity data sets were filtered
following the procedure described in Mollaret et al. (2019). Data inversion was conducted using the Software RES2DINV
(Loke, 2020) and typical inversion parameters used for heterogeneous and highly resistive terrain  (Hilbich et al., 2009;
Mollaret et al., 2019). Inversions with other schemes such as the open-source library pyGIMLi (Rücker et al., 2017) gave
comparable results (Mollaret et al., 2020).

**Table 2: Overview over all ERT and RST profiles in all regions, including classification into landform types, availability of ground truthing data and indication of permafrost presence or absence, where possible. Abbreviations: RG = rock glacier, TS = talus slope,**
**SED = sediment slopes (including gelifluction slopes, colluvial slopes, debris-covered bedrock, moraines, landslides).**



| | Profile | A01 | A02 | A03 | A04 | A05 | A06 | A07 | A08 | A09 | A15 | A16a | A16b | A17 | A24 | A25 | B01 | B02 | B03 | C02 | C03 | C04 | C06 | C07 | C08 | C09 | C10 |
|---|---|---|---|---|---|---|---|---|---|---|---|---|---|---|---|---|---|---|---|---|---|---|---|---|---|---|---|
| Meta | location | A | A | A | A | A | A | A | A | A | A | A | A | A | A | A | B | B | B | C | C | C | C | C | C | C | C |
| | landform | RG | RG | RG | RG | TS | PR/RG | RG | RG | RG | RG | PR | RG | RG | SED | TS/PR | RG | RG | RG | SED | SED | SED | SED | SED | SED | SED | SED |
| | altitude | 3762 | 3730 | 3742 | 3840 | 3850 | 3810 | 3830 | 3775 | 3785 | 3730 | 3708 | 3740 | 3650 | 3710 | 3695 | 3680 | 3800 | 3850 | 5085 | 5065 | 5005 | 5110 | 4910 | 4975 | 5080 | 4995 |
| ERT | length [m] | 144 | 595 | 144 | 235 | 213 | 213 | 188 | 380 | 94 | 141 | 94 | 188 | 285 | 213 | 141 | 355 | 213 | 285 | 142 | 94 | 69.5 | 94 | 142 | 94 | 94 | 141 |
| | spacing [m] | 3 | 5 | 3 | 5 | 3 | 3 | 4 | 4 | 2 | 3 | 2 | 4 / 2 | 3 | 3 | 3 | 5 | 3 | 3 | 2 | 2 | 1.5 | 2 | 2 | 2 | 2 | 3 |
| RST | length [m] | 69 | 164 | 69 | - | 180 | 164 | 164 | 164 | 69 | 69 | 82 | 164 | - | - | 46 | 123 | 177 | 123 | 46 | 46 | | | 103 | | | |
| | spacing [m] | 3 | 4 | 3 | - | 3 | 4 | 4 | 4 | 3 | 3 | 2 | 4 | - | - | 2 | 3 | 3 | 3 | 2 | 2 | | | 2.5 | | | |
| ground truth | borehole | | x | | x | | x | x | xx | | | x | x | | | | | | | | | | | | | | |
| | test pit | | x | x | | | | | xxx | x | | x | x | | | | | | | | | | | | x | x | x |
| | other | | | | | | | | | | | | | | | | | | | | | | | x | | | |
| | permafrost | y | y | y | y | ? | y | y | y | y | y | ? | y | n | n | ? | ? | y | y | y | ? | n | y | y | y | y | y |

| | Profile | D01 | D02 | D03 | D04 | D05 | D06 | D06b | D07 | D08 | D09 | F01 | E03_A | E03_BV | E03_BH | E04 | E05 | E08 | E09 | E11_H | E11_D | E12 | E13 | E14 | E15 | E16 | E17 |
|---|---|---|---|---|---|---|---|---|---|---|---|---|---|---|---|---|---|---|---|---|---|---|---|---|---|---|---|
| Meta | location | D | D | D | D | D | D | D | D | D | D | F | E | E | E | E | E | E | E | E | E | E | E | E | E | E | E |
| | landform | SED | SED | SED | SED | SED | SED | SED | SED | SED | SED | RG | SED | SED | SED | SED | SED | RG | RG | SED | SED | SED | SED | SED | SED | TS/PR | PR |
| | altitude | 5080 | 5135 | 5160 | 5105 | 5100 | 5025 | 5025 | 5030 | 5085 | 5108 | 4408 | 4520 | 4520 | 4520 | 4815 | 4815 | 4620 | 4640 | 4615 | 4615 | 4830 | 4545 | 4415 | 4400 | 4210 | 4210 |
| ERT | length [m] | 70.5 | 104 | 94 | 94 | 188 | 142 | 47 | 328 | 188 | 70.5 | 235 | 94 | 47 | 141 | 142 | 70.5 | 470 | 235 | 235 | 235 | 70.5 | 235 | 94 | 94 | 142 | 235 |
| | spacing [m] | 1.5 | 1.5 | 2 | 2 | 2 | 2 | 1 | 2 | 2 | 1.5 | 5 | 2 | 1 | 1.5 | 2 | 1.5 | 5 | 5 | 5 | 5 | 1.5 | 5 | 2 | 2 | 2 | 5 |
| RST | length [m] | | | | 82 | | | | | 46 | 46 | 69 | | | | | | 123 | | 69 | | | | | | | |
| | spacing [m] | | | | 2 | | | | | 2 | 2 | 3 | | | | | | 3 | | 3 | | | | | | | |
| ground truth | borehole | | | | | | | | | | | | | | | | | | | | | | | | | | |
| | test pit | | | | | | | | x | | | | xx | x | | x | x | | | | | | x | | x | | |
| | other | | | x | x | | | | | | | x | | | | | | | | | | | | | | | |
| | permafrost | y | y | y | y | y | y | y | y | y | y | y | n | n | n | ? | ? | y | y | ? | ? | y | n | n | n | y | y |

Refraction seismic data were recorded through a Geode system (Geometrics) with 24 geophones and a sledgehammer as source. First breaks were picked manually and afterwards inverted within the software REFLEXW (Sandmeier, 2020) to yield

tomograms of P-wave velocity on co-located lines of specific ERT profiles. The resolution and data quality differ for each profile and method; in general, the resulting root-mean-square errors of the ERT profiles were below 10 % (except for E17 with 22 %) and below 3 ms for the RST inversion (Table A1). See Table 2 for details on the individual profiles.

Regarding quantification of the volumetric ground ice content (ice content), Hauck et al. (2011) introduced a petrophysical approach by way of the so-called 4-phase model (4PM) using the obtained specific resistivity and P-wave velocity distributions

as input variables. The 4PM consists of a combination of two basic mixing rules for electrical resistivity (Archie's Law, Archie, 1942) and seismic P-wave velocities (a modified Wyllie equation, see Timur, 1968), and the condition that the volumetric contents of ice, water, air and rock sum up to 1 for each model cell. Under the assumption of a site-specific porosity distribution, the 4PM estimates the ice-, water and air content for each model cell. Wagner et al. (2019) extended the approach to a petrophysical joint inversion (PJI) model, which yields physically consistent estimates of all 4 phases, i.e. without the necessity

of prescribing porosity. Both model approaches were successfully applied to various permafrost occurrences (Halla et al., 2021; Mollaret et al., 2020; de Pasquale et al., 2020; Pellet et al., 2016; Schneider et al., 2013). However, the PJI still faces convergence problems in the absence of a priori knowledge, and its application to a large number of geologically and





geomorphologically different profiles is therefore challenging. Therefore, we opted here for the application of the 4PM, which allows consistent ice content modelling for a large number of profiles.

In the 4PM, the largest uncertainties in absolute ground ice content values are due to the absence of reliable porosity information and extreme values of pore water resistivities. The later are a factor in Archie's Law that must be prescribed (Hauck et al., 2011). Halla et al. (2021) established a procedure using ranges of porosity and pore water resistivity values to quantify the uncertainty in absolute volumetric ice content estimates of a rock glacier in the Argentinian Andes.

Within this study, we used ERT surveys to detect ground ice occurrences and delineate their vertical extent. As seismic surveys
are much more time-consuming, they were conducted only at specific ERT profiles to get quantitative ice content estimates at representative locations. As co-located ERT and RST profiles are necessary to provide input data for the 4PM, these model results are only available for 22 profiles (see Table 2). Ice content estimates and ground ice extent were estimated from ERT data alone for all other profiles. Hereby, resistivity averages and maxima were evaluated within so-called zones-of-interest (ZOI), i.e. the profile region, which is assumed to be representative for the landform and permafrost occurrence (Etzelmüller
et al., 2020). Validation data are available for several profiles and ZOI's through drill cores, borehole temperature information and test pits (see next section).

## 3   Study sites and data set

Between 2016 and 2019, five extensive geophysical campaigns were completed in three different regions of the Central Andes on both sides of the border between Chile and Argentina. In total 52 ERT and 24 RST profiles were acquired to characterize
permafrost conditions regarding extent, active layer thickness and ground ice content. All field data were acquired in the austral summer as part of characterising the periglacial environment. Profile locations were chosen according to the probable presence of frozen ground, but also according to easy access and safety. Apart from the fact that some of the considered permafrost landforms had surface disturbances (e.g., access roads or drilling platforms), the context of the projects has no further relevance for the scientific content of this paper. The available infrastructure, however, enabled access to high-altitude permafrost
environments and made possible the collection of a large and unique data set, including in-situ validation data.





Figure 2: Photographs of typical landforms with survey lines.





Due to the different locations (cf. Figure 1), a large variety of ground conditions ranging from sediment slopes (including
gelifluction slopes, colluvial slopes, debris-covered bedrock, moraines, landslides) over talus slopes, protalus ramparts (also
called protalus rock glaciers, or embryonic rock glaciers, cf. Barsch, 1996; Hedding, 2011) to rock glaciers is covered by
geophysical profiles. Table 1 summarises the main characteristics of the different study sites and geophysical profiles, and
Figure 2 shows some typical examples of the considered landforms with the geophysical profile lines indicated. Many of the
rock glaciers in the different investigation areas show initial or advanced signs of degradation (e.g., inactive front slopes,
thermokarst depressions), but in the absence of kinematic data for most of the observed rock glaciers a reliable determination
of their activity state according to the guidelines of the IPA action group on rock glacier inventory and kinematics (RGIK,
2020) remains challenging. As the activity of a rock glacier is not directly linked to its ice content, which is the focus of this
paper, we avoid any pre-classification of the rock glacier activity here, even if geomorphological indications and kinematic
data are available in some cases.

In total 24 coinciding ERT and RST profiles were subsequently used for the estimation of the ground ice content and its spatial
variability based on the 4PM. The availability of undisturbed core drillings, borehole temperature measurements and numerous
test pits enabled the validation of the methodological approach at 24 of the profile lines (availability of ground truthing data
indicated in Table 2).

## 4    Results

All available data (ERT/RST) have been quality-checked, processed and interpreted. An overview about data quality (filter
statistics, RMS error) and a reference plot with all available ERT and RST tomograms is provided in the Appendix (cf. Table
A1, Figures A1 and A2). In the following, we will present exemplary results regarding different landforms characteristics.

### 4.1    Rock Glaciers and Protalus Ramparts

#### 4.1.1    General characteristics

In total, 19 ERT profiles were measured on ice-rich permafrost landforms, including rock glaciers (16) and protalus ramparts
(3). These are shown in Figure 3 with the same dimensions and colour scales for all tomograms. All profiles have been analysed
and interpreted independently (Hauck et al., 2017; Hilbich et al., 2018; Hilbich and Hauck, 2018a, 2018b, 2019); here, we
focus on a general and comparative analysis of all profiles, as a detailed discussion of each case study is beyond the scope of
this paper.

Among our data, the resistivities of rock glaciers can be grouped into two parts: rock glaciers with resistivity maxima of the
permafrost body below the active layer well above 100 kΩm, partly reaching 1 MΩm or more (RG I, Figure 3a), and rock
glaciers with resistivity maxima mostly < 100 kΩm and/or shallower and more patchy resistive zones (RG II, Figure 3b). Rock





glaciers of group RG II often show visible degradation expressions, such as inactive front slopes or thermokarst depressions.

Protalus ramparts show similar resistivity values as rock glaciers (Figure 3c).

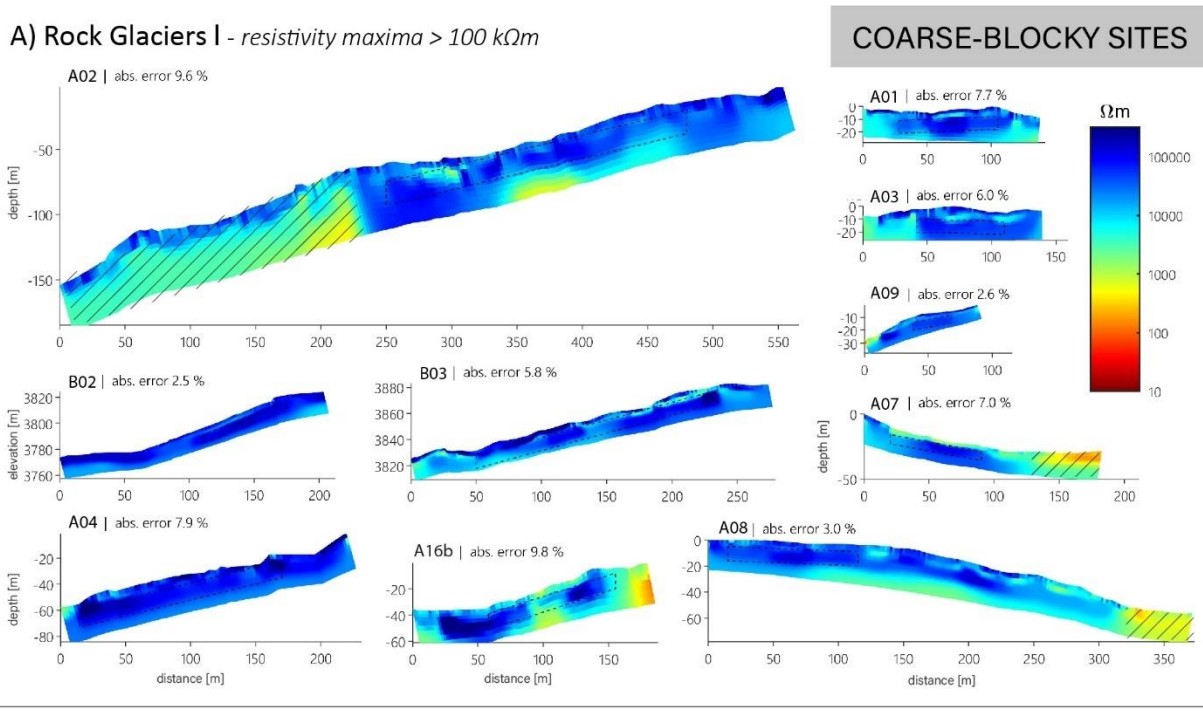

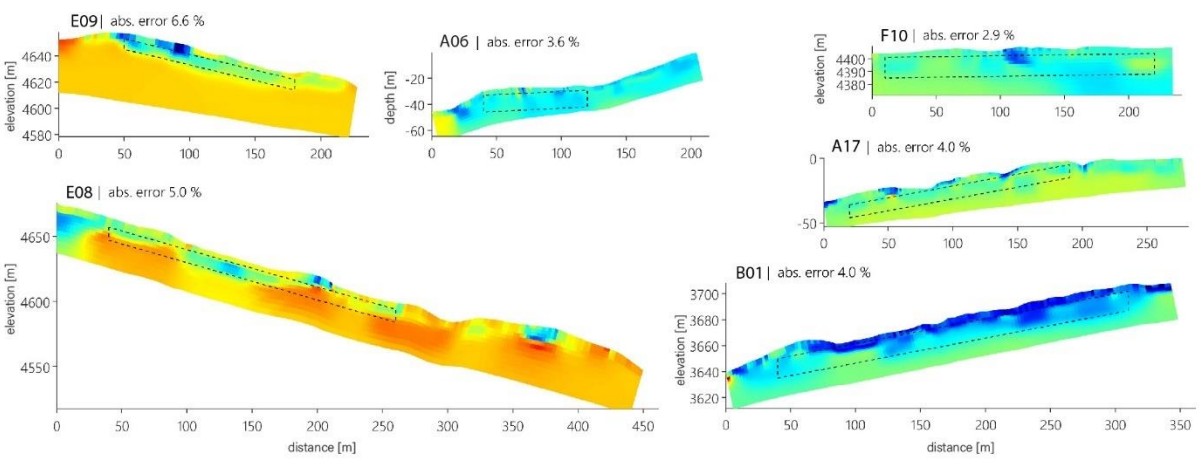

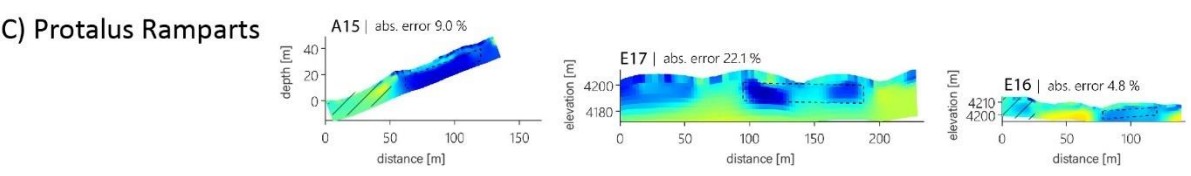





**Figure 3: Inverted ERT tomograms for all rock glacier and protalus rampart profiles of the study. Zones of the tomograms not related to a rock glacier or protalus rampart are indicated by diagonal lines. The dashed rectangles mark the so-called zone-of-interest (ZOI) used for the resistivity averaging and comparison. Data misfit (absolute error in %) is indicated for each profile.**

Note, that rock glaciers with a very coarse-blocky and dry active layer (with air-filled voids) typically can have similarly high resistivities in the active layer as in the ice-rich permafrost layer, as both air and massive ice are electrical isolators (e.g., profiles A04, A15). However, at the bottom of the active layer, more fine-grained material typically accumulates, and moisture from snowmelt and seasonal active layer thawing may be retained on top of the impermeable frozen layer, often resulting in a more conductive intermediate layer (e.g., visible in A01, A03, A08, A09, cf. Figure 3a).

A high-resistive zone indicating ice-rich permafrost can usually be observed throughout the entire landform for rock glaciers of group RG I, but with varying thicknesses and specific resistivity values. We estimate the thickness of the ice-rich permafrost body of the rock glaciers from the thickness of this high-resistive zone. As the resolution of geophysical methods generally decreases with depth, the determination of the upper boundary is more reliable than its vertical extent. The resolution of the lower boundary depends on several factors:

    a) survey geometry, defining the spatial resolution and the depth of investigation;

    b) the depth of the lower boundary in relation to the investigation depth (the shallower the boundary, the better its resolution); and

    c) the resistivity contrast between ice-rich permafrost layer and underlying layer (e.g., bedrock, the higher the contrast, the better the resolution).

We therefore use the onset of a decreasing resistivity gradient (below the maximum) as a conservative indicator for the lower limit of ice-rich permafrost.

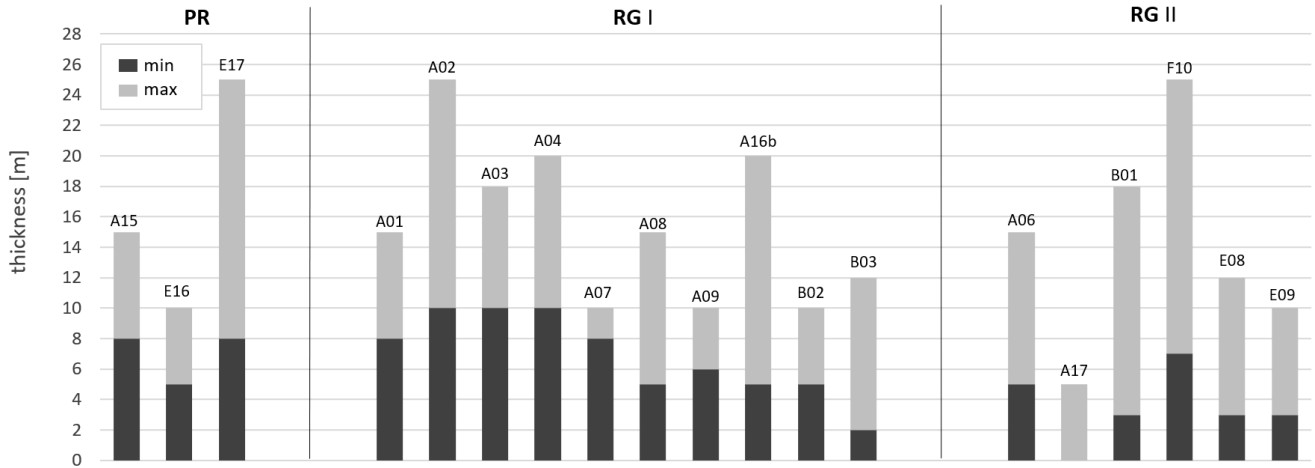

**Figure 4: Overview over minimum and maximum thickness of ice-rich permafrost in all protalus ramparts (PR), and rock glaciers (RG) of group I and II, determined from the high-resistive zone in the ERT tomograms. Profile A17 is from a relict rock glacier with little probability for ground ice.**





The investigation depth was sufficient to identify the bottom of the ice-rich permafrost layer for most rock glacier profiles. Due to the spatial heterogeneity within the observed profiles, the thickness of the ice-rich permafrost layer cannot reliably be determined everywhere along the profile and remains an estimate. Figure 4 indicates the ERT-based minimum and maximum
thickness of the ice-rich layer in all ice-rich permafrost profiles (i.e. rock glaciers and protalus ramparts). Note, that the minimum thickness refers to the ice-rich zones within the tomograms, and that most profiles also contain zones without permafrost or ice-rich layers. The determined thicknesses mainly range between a few meters and do not exceed 25 m for all considered landforms. No clear difference is observed for the different categories (and was not expected).

As an overall observation, it can be noted that data quality is often worse on the coarse-blocky parts of the rock glaciers because
of challenging conditions for sufficient galvanic coupling at the surface (Hilbich et al., 2009; Mollaret et al., 2019) than for the generally more fine-grained surface material and lower resistivities of rock glaciers with advanced degradation (RG II). This clearly affected the data quality in the first half of profile E17 (22 % data error, cf. Figure 3c), but had no severe impact on most other profiles (a few more profiles with insufficient data quality exist, but were not considered for this study).

### 4.1.2     Example data set: Rock glacier A16B

As an example, Figure 5 shows the geophysical results for profile A16B, which crosses two neighbouring rock glacier lobes, with a borehole drilled in one of the lobes marked by the black vertical line. The active layer was largely removed through the construction of the borehole platform. Maximum resistivities of up to 1 MΩm are observed in two distinct anomalies corresponding to the two different lobes, and indicate high ground ice content occurrences of 5-18 m thickness, which is confirmed by the drilling results (cf. Figure 5a, Figure 6, Table 3).

The corresponding seismic results (Figure 5b) confirm ice-rich permafrost with P-wave velocities of 3000-4000 m/s within the zone of the high resistivity anomalies. Below this zone, P-wave velocities of up to 6000 m/s indicate the bedrock at around 20 m depth. The profile clearly illustrates coinciding characteristic resistivity/velocity values for pure ice ($\rho > 1$ MΩm & $v_p = 3500$ m/s) and bedrock ($\rho \sim 1$ kΩm & $v_p = 6000$ m/s, cf. Hauck et al., 2011).

Based on the co-located ERT and seismic profiles, the volumetric fractions of the four phases rock, ice, water and air have
been modelled using the 4PM (cf. section 2). Figure 5c shows the modelled ice content for profile A16B, with two anomalies of > 60 % ground ice content, which is in good agreement with the previous interpretation and the results from the borehole stratigraphy. The thaw depth is around 3-5 m in both lobes (except for the disturbed area of the drilling platform).

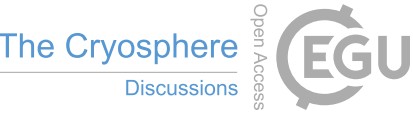



**Figure 5:** **Geophysical results of A16B (rock glacier): a) section of the ERT profile covered by the RST profile (see Figure 3a for full**
**profile), b) RST profile, and c) volumetric ice content modelled by the 4PM. The borehole position is indicated, with the light blue**
**part indicating the frozen part. The data misfit of both inversion models is indicated in the upper right corner. The cross-hatched**
**zone marks the area detected as bedrock in the 4PM. Labels denote the interpretation.**





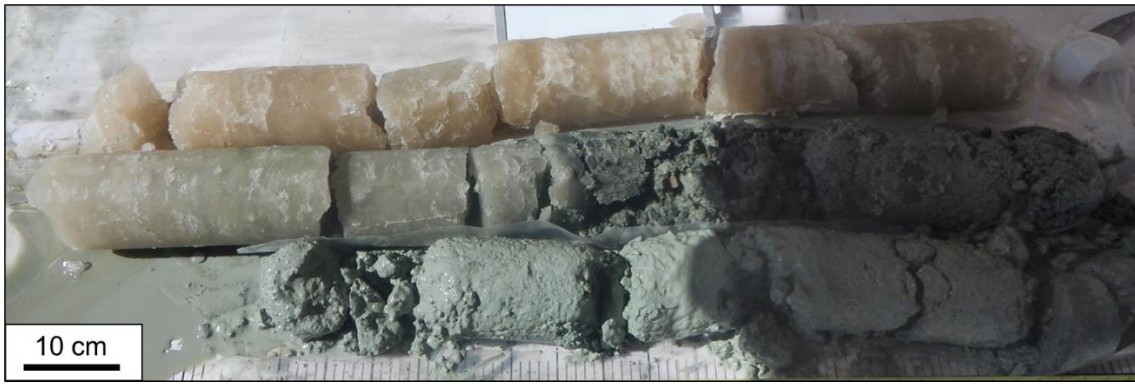

**Figure 6: Example of a frozen core extracted from a rock glacier at 11 – 14 m depth (ERT profile A16b). The upper half of the core run shows massive ground ice and the lower half frozen gravel and sand with a very low ground ice content.**

**Table 3: Overview over available ground truthing data with the most important permafrost-relevant information. Abbreviations: BH = borehole; TP = test pit; PF = permafrost; AL = active layer. The presence of one or more boreholes or test pits is indicated with one or more 'x', similarly one or more 'y' or 'n' indicate, if permafrost was confirmed or not. The ice content range represents the observed minimum and maximum values throughout the entire depth column.**

| Profile | type of ground truthing data | | | | type of validation | | | | comments |
|---|---|---|---|---|---|---|---|---|---|
| | BH | TP | natural outcrop | max. depth | PF confirmed | AL thickness | PF thickness | ice content | |
| A02 | x | xxx | | 28 m | yyy | 3.2 m | 25 m | 45 - 90 % | |
| A04 | x | | | 32 m | y | 2 m | 21 m | 25 - 75 % | |
| A06 | x | | | 31 m | y | 8 m | 12 m | 0 - 95 % | |
| A07 | x | | | 29 m | y | unknown | 25 m | 40 - 85 % | |
| A08 | xx | xx | | 25 m | y | 2 m | > 15 m | 10 - 100 % | Monnier & Kinnard (2013) |
| A16a | | x | | 5.1 m | n | | | | minimum 5.1 m unfrozen, TP infront of the protalus rampart |
| A16b | x | x | | 15.4 m | y | unknown | 13 m | 70 - 100 % | AL removed during drilling |
| A17 | x | | | 37 m | n | | | | minimum 37 m unfrozen |
| C06 | | | x | 2 m | y | ~0.5 m | | unknown | nearby test pits indicate shallow PF (< 0.5 m) with 60 - 75 % ice content |
| C07 | | x | | 1.3 m | y | 1.2 m | | 40 % | ice-rich sediment |
| C08 | | x | | 1.3 m | y | 1.2 m | | 100 % | > 10 cm thick ice lens |
| C09 | | x | | 0.9 m | y | 0.8 m | | 70 % | 10 cm thick ice lens with 70 % ice content |
| D05 | | | x | 1.5 m | y | ~0.5 m | | unknown | variable ice content, incl. ice lenses with 100 % |
| D07 | | x | | 5 m | y | ~0.5 m | > 5 m | 10 - 50 % | variable ice content, incl. ice lenses with 100 %; liquid water flow observed within the PF layer at 3-4 m depth |
| D09 | | | x | 5 m | y | unknown | | 60 - 70 % | ice-rich bedrock |
| E03_A | | xx | | 6.5 m | nn | | | | minimum 6.5 m unfrozen |
| E03_BV | | xx | | 7.3 m | nn | | | | minimum 7.3 m unfrozen |
| E04 | | x | | 2.2 m | unclear | ~2.1 m | | 0 % | potentially frozen at >2.1 m depth, no ice visible |
| E05 | | x | | 4.3 m | y | ~0.4 m | | 0 % | negative temperature, no ice visible |
| E12 | | x | | 4.6 m | y | ~0.25 m | | < 5 % | frozen layer with ice-coated rocks (ice-poor) |
| E15 | | x | | 8 m | n | | | | minimum 8 m unfrozen |



### 4.2 Talus slopes

#### 4.2.1 General characteristics

ERT profiles were collected on three talus slopes, and all of them show a similar resistivity pattern: a layer of increased
resistivity (~10 kΩm) within the talus material having a bulk resistivity of only a few kΩm (Figure 7). The resistive layer is
located at depths > 3 m, i.e. below a potential active layer, and has a maximum thickness of 10 m for the four measured profiles.
The resistivities are sufficiently high to support the hypothesis of frozen conditions within the talus slope (Hauck and Kneisel,
2008), even if the expected ground ice content would be low. The resistive zone could also be explained by purely air-filled
voids within the porous coarse-blocky substrate, similar to the resistive anomalies visible directly at the surface in most
profiles. This ambiguity can, in general, be addressed through coinciding seismic profiles (available for A05, A16a, and A25)
and will be shown exemplary in section 4.2.2. Unfortunately, no ground truthing information is available for any of the talus
slopes.

#### 4.2.2 Example data set: Talus slope A05

Profile A05 is a longitudinal profile within a talus slope, located on an east-facing slope in the western part of a valley with
numerous rock glaciers at its south- and west-facing slopes. The ERT results in Figure 8a show comparatively low resistivities
of < 10 kΩm in most parts of the profile, indicating no or very small ground ice content. A localised anomaly with higher
resistivities ($\rho \geq 10$ kΩm) exists between 80 - 140 m horizontal distance and suggests a small possibility for potential ground
ice at approximately 5 - 12 m depth. Seismic velocities of $v_p < 1500$ m/s in the same region point to loose blocks and debris
with air-filled voids (Fig. 8c), rather than a layer with massive ground ice, except at larger depths (~25 m), where higher P-
wave velocities ($v_p \sim$ 2000-3000 m/s) and coinciding low resistivities ($\rho < 5$ kΩm) strongly indicate the bedrock. No ground
truthing data are available for this profile. The anomaly with slightly larger resistivity values around 10 kΩm between distances
80 - 140 m could indeed indicate frozen conditions, but with volumetric ice contents, which are too small to be detected by
our seismic survey set-up. Consequently, the 4PM-estimated ice content is close to zero within the whole model domain (Fig.
8d).





**Figure 7: Inverted ERT tomograms for all talus slope profiles of the study. Zones of the tomograms not related to the talus slope are indicated by diagonal lines. The dashed rectangles mark the so-called zone-of-interest (ZOI) used for the resistivity averaging and comparison. Data misfit (absolute error in %) is indicated for each profile.**





**Figure 8: Geophysical results of A05 (talus slope): a) section of the ERT profile covered by the RST profile (see Figure 7 for full profile), b) RST profile, and c) volumetric ice content modelled by the 4PM. The data misfit of both inversion models is indicated in the upper right corner. Labels denote the interpretation.**






### 4.3    Sediment slopes

#### 4.3.1    General characteristics

In addition to the 22 profiles on pebbly and coarse-blocky substrates of rock glaciers, protalus ramparts and talus slopes, 30 additional ERT profiles were measured in more fine-grained sedimentary substrate, including colluvial slopes (17 profiles),

gelifluction slopes (4 profiles), and weathered bedrock covered with a shallow debris layer (9 profiles, cf. Figure A2). Some of these ERT profiles on sediment slopes do not contain permafrost (e.g., E03, E13, E14, E15, cf. Table 2). However, all profiles on sediment slopes show significantly lower resistivities (mostly well below 1 kΩm) compared to rock glaciers, protalus ramparts and talus slopes (cf. Figure A1), including those, where ground ice was confirmed by test pits or outcrops (e.g., D07, D09, C06, C08). The reduced resistivity values are a result of the fine-grained and partly humid substrate and/or

the weathered bedrock, as well as the generally lower volumetric ice content in sediment slopes in the form of interstitial ice (i.e. << 50 %, except for excess ice in mostly thin ice lenses).

In addition, many of these profiles contain prominent conductive layers of < 100 Ωm (Figure A2). We speculate that this is mainly caused by (a) conductive sediments stemming from eroded hydrothermally altered bedrock, which was transported downslope (in case of the colluvial slopes), (b) the altered/conductive bedrock itself, or partly also (c) liquid (supercooled)

water due to freezing point depression by increased ion content related to hydrothermal alteration (Hauck et al., 2017; Hilbich and Hauck, 2018a). Significant water flow was for example observed above an ice-rich layer in a 4-5 m deep test pit close to profile D07 and below a frozen layer in profile C08 (cf. Figure A2a). It is important to note that these conductive layers are with high probability features strongly amplified by the inversion process caused by preferential current flow through conductive layers, which strongly biases the inversion result towards this conductive layer. Various synthetic modelling studies

(e.g., Hilbich et al., 2009; Mewes et al., 2017) have shown that the real thickness of such conductive layers may be more than an order of magnitude smaller than illustrated in the resulting tomograms. In this case, the resistivity of layers below will be biased towards lower values (i.e. ice contents may well be higher than expected from the inverted values), while the depth of the deeper layers could be strongly overestimated.

The detection of permafrost occurrences is further complicated by the often thin or patchy ice-lenses, which cannot be detected

with confidence because of the trade-off in the electrode spacing between reasonable large investigation depth/profile length and the resulting reduced spatial resolution capacity. A reliable interpretation of these tomograms is therefore not straightforward, but experience from the synthetic modelling studies mentioned above and comparison with ground truthing information allows resistive anomalies caused by small ice lenses to be identified, even if absolute resistivity values are lower than commonly known to indicate frozen conditions. Similar cases are known from the European Alps, where the combination

of low-resistive geologic host material, increased water content and temperatures close to the freezing point leads to similarly low permafrost resistivities (Hilbich et al., 2008; Mollaret et al., 2019; Noetzli et al., 2019).

Permafrost was clearly detected in profiles C07, D04, D05 and D09, where a prominent resistive layer (> 10 kΩm) is observed. Similar values but within much smaller and thinner anomalies were found in profiles D06, D07, D08. Test pits and natural outcrops within incised channels confirm the presence of permafrost for profiles D04 - D07 and D09 (cf. Figure A2).






### 4.3.2    Example data set: Colluvial Slope D04

Figure 9 shows the results of profile D04 located within an east-facing slope, and consisting of mainly fine-grained colluvial sediments, cut by incised channels, which are active during snow melt (cf. Figure 2e). The slope shows a slightly convex form, indicating the potential for ice-rich conditions. The ERT tomogram in Figure 9a shows a high-resistive layer with values up to

10 kΩm indicating ice-rich permafrost between approximately 2 - 8 m depth, with a maximum around 3 m depth. Maximum resistivity values are similar to the maximum values in profile D09, where ground truthing from a test pit confirmed ice contents ≥ 50 %. We therefore expect significant ice-rich layer(s) in this profile with a possibly supersaturated zone around the resistivity maximum. The presence of ice-rich sediments is further confirmed by a natural outcrop formed by an incised channel close to profile D04, which exposed ice-rich and partly supersaturated sediments at about 1 m depth (thickness of this layer

unknown).





**Figure 9: Geophysical results of D04 (colluvial sediments): a) section of the ERT profile covered by the RST profile (see A2a for full profile), b) RST profile, and c) volumetric ice content modelled by the 4PM. The data misfit of both inversion models is indicated in the upper right corner. The cross-hatched zone marks the area detected as bedrock in the 4PM. Labels denote the interpretation.**


Below this ice-rich layer, resistivities < 500 Ωm prove ice-poor conditions, while a reliable differentiation between sediment or bedrock is not possible without further information. Seismic P-wave velocities increase to > 5000 m/s at ~10 m depth and indicate a transition to more competent frozen rock at this depth (Figure 9b).





Figure 9c shows the modelled ground ice content with maximum values around 60 %, which confirms the expected high
ground ice contents in this profile. Highest values are observed between 30 - 70 m horizontal distance with decreasing values
in upslope direction (40 - 50 %) and an abrupt change to values < 30 % near the road in the lower part of the profile.

## 4.4    Joint analysis

### 4.4.1    Mean resistivity and P-wave velocity

Comparing mean resistivity and P-wave velocities of the various profiles is a delicate task due to their dependence on
(potentially very different) local geologic conditions, which may give rise to substrate-dependent resistivity/velocity variations
that may be misinterpreted as differences in ground ice content. Besides, uncertainties due to different measurement
configurations and inversion errors may further impact a joint comparison. On the other hand, the dependence of resistivity
and P-wave velocity on ice content is very strong, and its signal should be clearly detectable in such a large and comparatively
homogeneous dataset presented in this study.

For a joint analysis of the representativeness of the measured geophysical parameters for the considered landforms, we selected
all profiles where the presence of permafrost a) has been identified, or b) is considered possible but not confirmed (e.g. in talus
slopes, cf. Table 2). We defined a rectangular zone either within the presumed permafrost occurrence (representative
permafrost zone), or within the zone most probable/indicative for permafrost in case of ambiguous interpretation and call this
zone hereafter the zone of interest (ZOI). The ZOI was chosen to be representative for the confirmed (or unconfirmed possible)
permafrost occurrence within the respective landform with minimal bias from potential inversion artefacts, thus resulting in
various sizes and positions of the ZOI for different profiles.

Mean and maximum resistivities/velocities within the ZOIs were then extracted, and they clearly show different resistivity and
velocity regimes for different landforms and substrates (see Figure 10a,b). Figure 10c analyses the relationship between mean
specific resistivity and P-wave velocity within the ZOI of co-located ERT and seismic profiles, and reveals a landform-specific
clustering of resistivity-velocity pairs. Hereby, the resistivity/velocity pairs of rock glaciers cluster in two parts (green and
purple in Fig. 10c). The purple cluster (lower resistivity and P-wave velocity) is consistent with the rock glaciers of group RG
II (see section 4.1.1), showing visible signs of advanced degradation and lower ground ice contents than the ones in the green
cluster (RG I). Similarly, lower velocity mean values are present for protalus ramparts (PR) and talus slopes (TS), with the
exception that TS show lower maximum resistivities than PR and RG II; probably due to their lesser ground ice contents.
While the two rock glacier groups clearly differ in their mean resistivities and velocities, their maximum values overlap
probably because most degrading rock glaciers still contain ice-rich zones with similar values as in intact rock glaciers (Figure
10a,b). Sediment slopes (often reaching bedrock at shallow depths) have clearly differing characteristics from coarse-blocky
sites, which is attributed to their lower porosity (higher velocity) and lower ground ice content (lower resistivity). Note,
however, that Figure 10c only provides an incomplete picture biased towards ice-rich landforms, as seismic surveys have
mainly been conducted on ERT profiles indicating ice-rich permafrost (cf. Table 2).



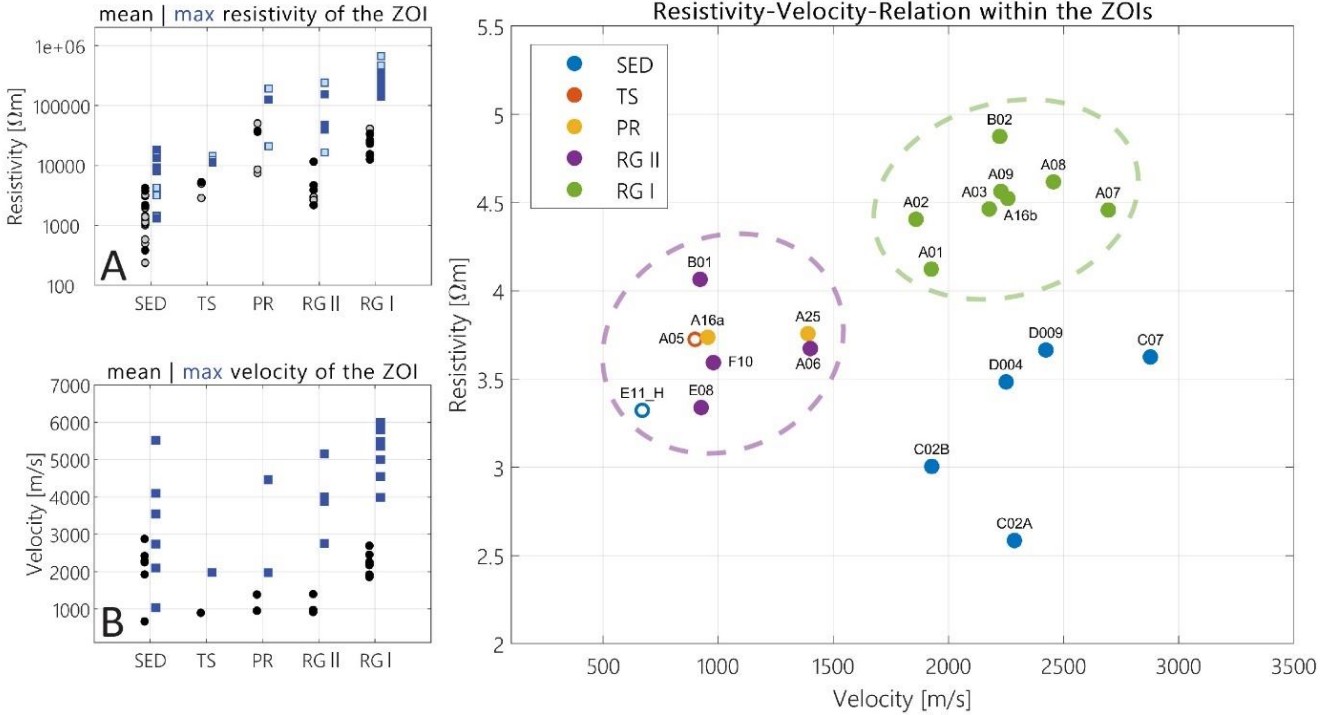

**Figure 10: Landform-specific distribution of mean (black) and maximum (blue) values of inverted a) resistivity and b) velocity within the ZOIs of the respective ERT and RST tomograms. c) Scatter plot of mean resistivity and velocity for all co-located ERT and RST profiles, classified after landforms. Unfilled symbols in a) denote ERT profiles without co-located RST profiles. Unfilled symbols in c) denote ZOIs with only possible permafrost occurrence. Abbreviations: SED = sediment slopes (including debris-covered bedrock, colluvial slopes, gelifluction slopes, etc.); TS = talus slopes; RG = rock glaciers (groups I and II described in section 4.1.1).**

The striking pattern in Figure 10c, with clustered and high resistivities for intermediate velocities (RG I), and low-intermediate resistivities for similarly high or even higher P-wave velocities (bedrock) is apparent and has already been noted by Hauck et al. (2007) for geophysical surveys on several permafrost landforms in the South Shetland Islands/Maritime Antarctica. Rock glaciers with massive ice cause maximum resistivities, but P-wave velocities around 3500 m/s, close to the literature value for ice. Sites with $v_p > 4000$ m/s usually indicate the presence of (unfrozen or frozen) bedrock, coinciding with lower resistivities due to the lower ice content. Mean seismic velocities in Figure 10b are all < 3000 m/s, which is certainly influenced by the limited investigation depth on some rock glaciers (bedrock not reached), but also represents the generally lower P-wave velocities of hydrothermally altered bedrock in some cases.

The systematic pattern observed in Figure 10 with a high consistency in the resistivity values over so many different surveys justifies the applicability of the geophysical approach to characterise different permafrost landforms, even in the absence of ground truthing. The seismic results further support and confirm the interpretation of the ERT data, but with a reduced overall representativeness due to a biased profile selection, fewer profiles and smaller profile dimension.

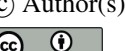


### 4.4.2 Volumetric ground ice contents

Similarly to Figure 4 (section 4.1.1), Figure 11a shows the estimated minimum (dark grey) and maximum (light grey) thicknesses of the ground ice layer for all profiles, where permafrost is a) confirmed or probable (indicated by blue frames), or b) unconfirmed and uncertain, but possible. The permafrost base for non-rock glacier sites could not always be detected, in these cases the base of the surficial ice-rich layer was determined and is plotted instead. It is clear that ice-rich layers in sediments are much thinner than in coarse-blocky substrates. Further, most ice-rich layers within our study are thinner than 25 m, including all rock glacier profiles. Quantitative model results for volumetric ice content (as presented exemplarily above) are available for a total of 21 profiles with co-located ERT and seismic surveys, including 12 rock glaciers, 2 protalus ramparts, 1 talus slope, and 6 sediment slopes. Figure 11b shows the modelled mean (dark grey) and maximum (light grey) volumetric ground ice contents within the defined ZOIs for all these profiles. The error bars give the uncertainty resulting from different 4PM runs spanning over the most probable porosity range for the respective landforms (SED: 30-45-60 %; TS: 40-50-60 %; RG: 40-60-80 %). The results indicate that maximum ice contents within the considered ZOIs are 51 - 56 % (+- 20 %), highest for rock glaciers of group RG I, and between 25 - 49 % (+- 20 %) for all other profiles. Note that anomalies with even higher ice contents can be present, but cannot explicitly be delineated if their size is smaller than detectable by the measurement configuration. More representative for the landform scale is, however, the mean ground ice content within the considered ZOIs, which spans in the same order of magnitude for most considered profiles (11 - 40 %) and shows that the ice content in the ice-rich layers of sediment slopes can be comparable to those of rock glaciers, even if the overall dimension of the ice-rich layer is very different.

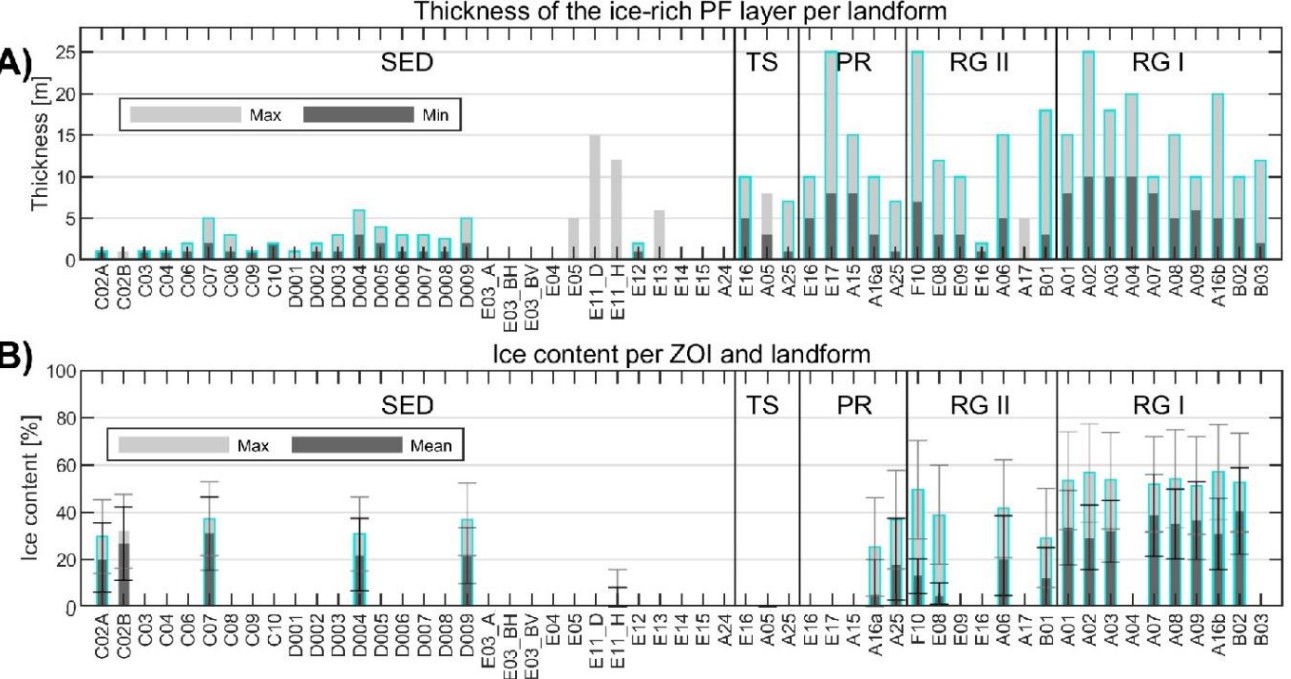





**Figure 11: Landform-specific distribution of a) minimum and maximum thickness of the ground ice layer, and b) the mean and maximum volumetric ground ice content within the ZOIs defined before, as derived from the 4PM. The bars and error bars in b) are based on landform-specific porosity ranges indicated in the text. Abbreviations: SED = sediment slopes (including debris-covered bedrock, colluvial slopes, gelifluction slopes, etc.); TS = talus slopes; PR = protalus ramparts; RG = rock glaciers. Profiles with confirmed or probable permafrost occurrence are highlighted in light blue, while the other values are only based on zones**
**with possible permafrost occurrence.**

## 5    Discussion

The uncertainty of the ice content estimation presented above depends first of all on the standard uncertainties of the geophysical data such as measurement data quality, resolution capacity, investigation depth, potential inversion artefacts, and representativeness of the geophysical profile for the whole landform. In addition, the uncertainties of the 4PM approach (rock-

ice ambiguity, porosity range, Archie parameter and estimate of rock P-wave velocity) have to be taken into account. In the context of mountain permafrost studies, 4PM-related uncertainties have already been addressed by Mewes et al. (2017) and Halla et al. (2021). In this study, we make additional use of the opportunity to compare our estimates with available ground truthing information, wherever possible, which when used as calibration reduces the uncertainty considerably. However, a large uncertainty remains regarding the representativeness of the individual profiles for a given landform. Depending on the

local geomorphological setting, ground ice contents can vary strongly, especially in case of very large landforms (e.g., Halla et al., 2021).

### 5.1    Comparison of results with ground truthing information

Since permafrost is thermally and temporally defined  (Muller, 1943) and can be present in different substrates under various porosity and saturation conditions, it can exhibit a wide range of possible values in the geophysical parameters. Attribution of

absolute electrical resistivity and P-wave velocity values to permafrost presence and specific ice contents can therefore be ambiguous without additional information. Further, the inverted geophysical parameters within a tomogram are influenced by the resolution capacity of the survey geometry in relation to the observed structure, the data quality and the material contrasts, which may all lead to inversion artefacts (Day-Lewis et al., 2005; Hilbich et al., 2009; Mewes et al., 2017). Small-scale anomalies and thin ice layers may not become visible in the comparatively coarse survey geometries utilised in the majority

of the profiles of our study.

Table 3 gives an overview over the different types of available ground truthing data (boreholes, test pits, natural outcrops), the respective depth range covered, and the type of validation provided by the different data.

In general, the interpretation of the tomograms (regarding presence/absence of ice-rich permafrost, cf. profiles highlighted in blue in Figure 11a) as well as the overall dimension of the active layer thickness (cf. ERT tomograms in Figure A1, A2) is

confirmed by the ground truthing data, thus enabling the spatial analysis of ground ice occurrence and its quantification.

For some rock glaciers (A02, A06, A07, A08, A16b), borehole-derived ice content values (representing minimum and maximum values observed throughout the borehole) can be compared to 4PM-derived min/max ice contents within the pre-



defined ZOIs (Figure 12). As thin ice-rich layers can be resolved by direct observations from boreholes or test pits but not necessarily by the relatively coarse survey geometries of the geophysical profiles, maximum ice content values observed in

the drill cores are generally higher. In addition, the 4PM cannot model super-saturated conditions (i.e. ice contents exceeding the assumed porosity), which further implies a bias towards underestimated maximum ice contents for the applied porosity ranges (cf. section 4.2.2). It is therefore not surprising, that the borehole-derived ice contents are mostly higher than the 4PM-derived values. Where quantitative ground truthing information is available, the 4PM can be calibrated by minimising the difference between the estimate and the ground truth, resulting in more consistent ice content values, as illustrated exemplarily

in Figure 13 for a profile with intermediate ice content (A02) and one with high ice content (A16b). Figure 13 further shows, that the porosity models of 60 or 80 % lead to more realistic ice content values than the lower-bound porosity model of 40 %. However, borehole validation provides highly valuable information on the point scale, but a direct comparison of borehole- and 4PM-derived ground ice contents remains challenging due to the different resolution capacities, dimensions (1-D vs. 2-D) and 4PM-related limitations. In the absence of such calibration data, ice content estimates of ice-rich permafrost layers may

be underestimated (as a consequence of underestimated porosity ranges) and rather represent lower-bound estimates. This bias is, however, also a direct consequence of the spatially averaging ZOIs, which also include zones with higher spatial variability and therefore smaller ice contents. On the contrary, boreholes represent single-point information and are usually placed where the maximum ground ice content is assumed.

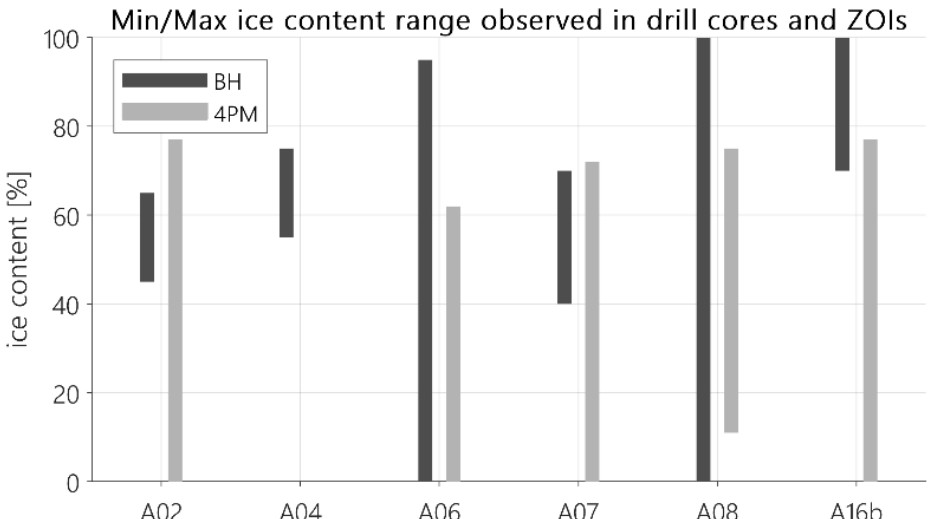

**Figure 12: Comparison of borehole-derived with 4PM-derived min/max ice contents within the pre-defined ZOIs. Here, only borehole values for the depth range covered by the ZOIs are considered.**

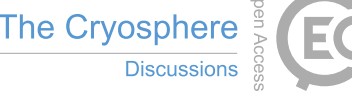



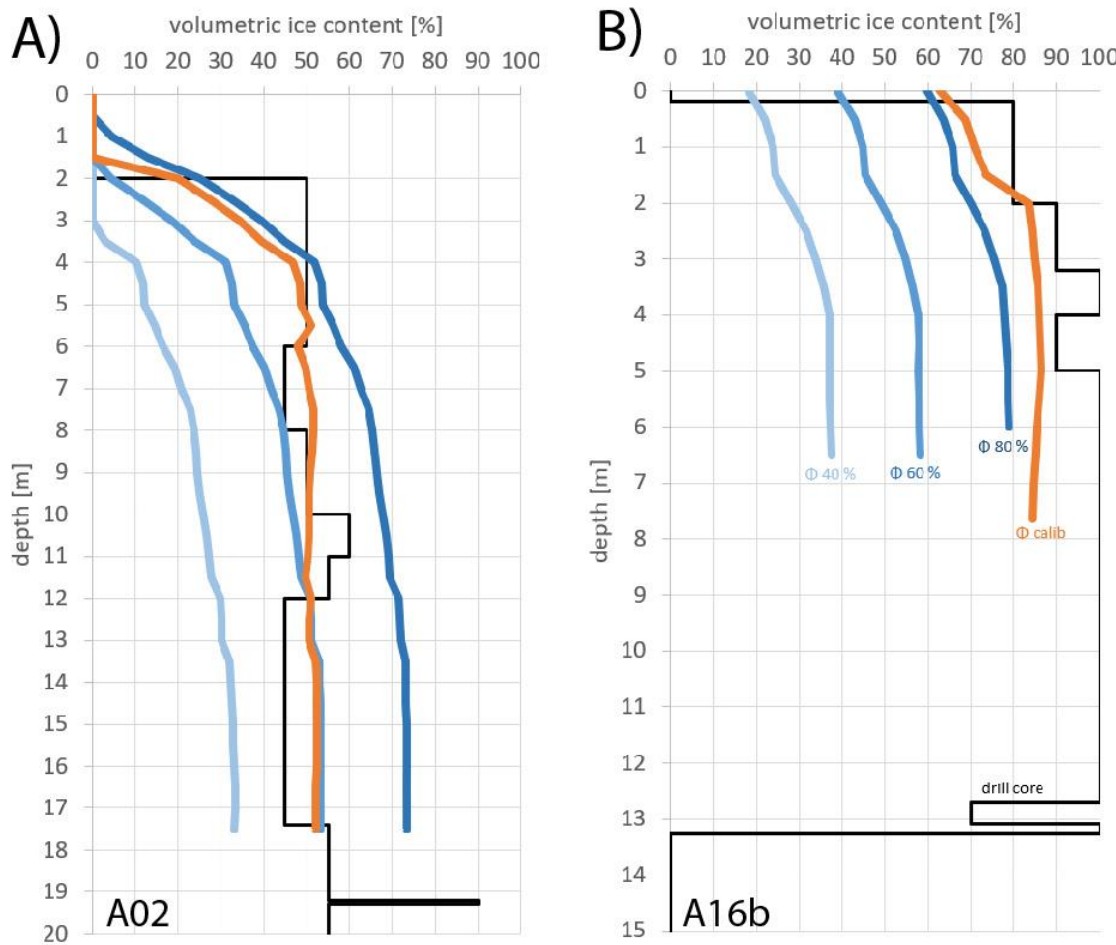

**Figure 13: Comparison of ground truthing information from a drill core (black line) with 4PM-derived ground ice volumes at the borehole position of profiles A02 (A) and A16b (B). The blue lines correspond to 4PM runs based on homogeneous porosity models with 40, 60, and 80 % (as used for the ZOIs), and the orange line shows the 4PM result after calibration with ground truthing data.**

### 5.2    Ice content of rock glaciers

Ground ice is present in the majority of all profiles, with ice contents ranging from a few percent by volume to clearly supersaturated conditions within various rock glaciers (Figure 3, Table 3). At sites with shallow sediment cover, small ice lenses are frequently present, which appear in the tomograms in the form of local resistive anomalies (cf. Table 3 and Fig. A2), and could be validated through various test pits and natural outcrops. Based on the estimates drawn from the 4PM simulations (considering the 60 % and 80 % porosity models), the rock glaciers with resistivity maxima > 100 kΩm (RG I) within our study areas show on average ground ice contents between 35 and 55 % by volume and thicknesses of the ice-rich layer of 3 to 25 m, but with a considerable spatial heterogeneity (cf. min/max estimates for the thickness of the ice-rich layer in Figure 11a, or the example in Figure 5). Our results further suggest, that the detected maximum ice contents within the ZOIs (35 - 75 %) roughly correspond to the general assumption on average ice contents within active rock glaciers found in the literature (40 –



60 %, cf. Arenson and Springman, 2005; Barsch, 1996), which implies, however, that this assumption may tend to overestimate mean ground ice contents on a landform scale. Care has therefore to be taken regarding general up-scaling approaches for quantitative estimates of the total ground ice content within a rock glacier. Several studies of the hydrologic role of rock glaciers in the Andes used an estimate of 50 % volumetric ice content as mean value for rock glacier bodies (e.g., Brenning, 490  2005; Perucca and Angillieri, 2011; Rangecroft et al., 2015). This commonly used estimate is often justified by borehole core data from rock glaciers elsewhere (e.g., Haeberli et al., 1988; Mühll and Holub, 1992). However, boreholes are usually drilled at promising locations for massive ground ice occurrences and the recovery of undisturbed samples with high ice contents is easier than sampling ice-poor samples. Therefore, results from boreholes are often biased towards ice-rich conditions, hence, do not represent mean conditions for the entire landform. Estimates of volumetric ice content using a homogeneous value of 495  50 % can therefore easily lead to over-estimations.

In addition, published estimates of total ground ice volumes within rock glaciers have been based on simplified relations between the surface area and average rock glacier thickness (i.e. area-thickness relations introduced by Brenning, 2005). Figure 14 compares the area-thickness estimates according to the approach by Brenning (2005) with our geophysics-based estimates for the rock glaciers of our study. This comparison suggests that the thickness of the ice-rich permafrost layer as inferred from 500  geophysical data is in most cases considerably smaller than the one approximated from commonly applied area-thickness-relations (cf. Azócar and Brenning, 2010; Janke et al., 2017; Rangecroft et al., 2015). Only for few of the very ice-rich landforms (e.g. E17 or A16b) the two approaches show comparable results. In addition, areal extents of rock glaciers are often not clear and very difficult to determine (Brardinoni et al., 2019; RGIK, 2020), especially in the case of complex landforms combining multiple rock glacier generations, resulting in a significant source of error when applying any rock glacier area- 505  thickness correlation.

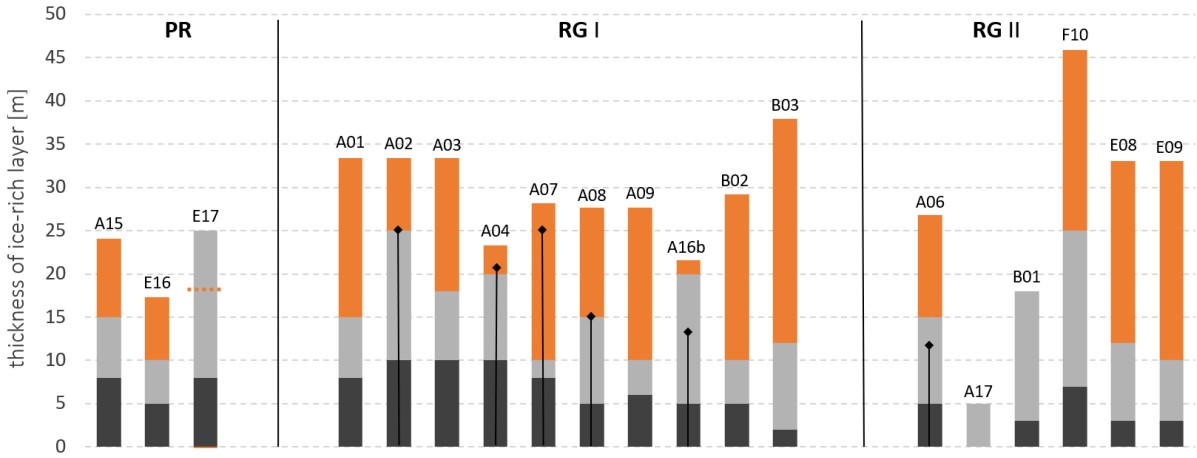

**Figure 14: Comparison of the minimum/maximum (black/grey) thickness of the ice-rich layer of rock glaciers (as in Figure 4) with the estimated rock glacier thickness according the area-thickness relation after Brenning 2005 (orange) for selected rock glaciers and protalus ramparts. Brennings approach has not been applied to A17 and B01, which are strongly degraded or relict rock glaciers. For E17, the area-thickness relation revealed a slightly smaller thickness (dashed orange line) than the geophysical maximum value. Black vertical lines illustrate the permafrost thickness revealed from borehole evidence, where available.**



Although previous assumptions of ground ice content within rock glaciers (40 - 60 %, e.g., Brenning, 2005) roughly correspond to our field-based results, this is only true for their ice-rich zone. As rock glacier bodies also consist of zones with considerably smaller ice contents (cf. Fig. A1), large-scale model studies using the above-mentioned area-thickness relations will introduce
a bias towards overestimation of total ice content with respect to total area. In the companion paper in part II, Mathys et al. (n.d.) propose a new upscaling approach of geophysically-based estimates of the ice volume per landform, which is compared to standard approaches using area-thickness scaling and constant ground ice contents per rock glacier. Similar to our results presented in Figure 14 they find lower total ground ice volumes in rock glaciers when estimates are based on geophysical data in the field compared to simplified rock glacier – ice content relations.

**5.3    Ice content of other landforms**

In contrast to remote-sensing-based approaches, which can only delineate rock glaciers as indirect representations of permafrost bodies with unknown relevance for the hydrological cycle (Azócar and Brenning, 2010), the geophysically-based approach presented in this study is not restricted to rock glaciers, but allows the estimation of ground ice content in a variety of landforms that constitute the periglacial belt. Examples are given in sections 4.2 and 4.3. Neglecting landforms other than
rock glaciers in most studies is due to the invisibility of their ground ice content from space (and during site visits) and the corresponding difficulties in obtaining field data from remote areas. Rough approximations indicate that even thin ice-rich layers in permafrost slopes at high elevations (e.g. Figure 9) may add up to similar ice volumes per catchment as present in catchments in zones where individual rock glaciers are present and only a medium probability of permafrost exists.

To investigate this hypothesis, we exemplarily upscaled the geophysics-based ice content estimates to the landform scale for
two sites, where ground truthing data is available. Based on geophysical results from six different profiles on a sediment slope (D03, D04, D05, D06, D07, D08), and three different profiles from a rock glacier (A01, A02, A03), the average thickness and ice content of the ice-rich layer of both landforms was approximated in terms of a lower-bound and an upper-bound estimate. Figure 15 shows the two landforms, the lower- and upper-bound estimates of the thickness of the ice-rich layer, as well as the estimated total ground ice volumes for the sediment slope and the rock glacier. The area of the rock glacier is approximately
0.11 km$^2$, which is about ten times smaller than the considered area of the colluvial slope (~1 km$^2$), but the rock glacier is expected to have a substantially thicker ice-rich layer of 10 - 15 m, compared to 0.5 – 1.5 m for the sediment slope. Assuming an average volumetric ice content of 50 % for the ice-rich layer at both sites leads to an approximated lower-bound estimate of the total ice volume of 250 000 m$^3$ for the sediment slope and 550 000 m$^3$ for the rock glacier. Considering the upper-bound estimate, i.e. the upper-bound average value for the thickness of the ice-rich layer (as opposed to its maximum within the
landform), estimated volumes range with 750 000 m$^3$ for the sediment slope and 825 000 m$^3$ for the rock glacier in the same order of magnitude. This indicates that even thin ice layers in sediment slopes can contain similar dimensions of ice volume per catchment as rock-glacier-dominated catchments. A more detailed analysis of this hypothesis using a newly developed upscaling approach is presented and discussed in the companion paper, part II (Mathys et al., n.d.).





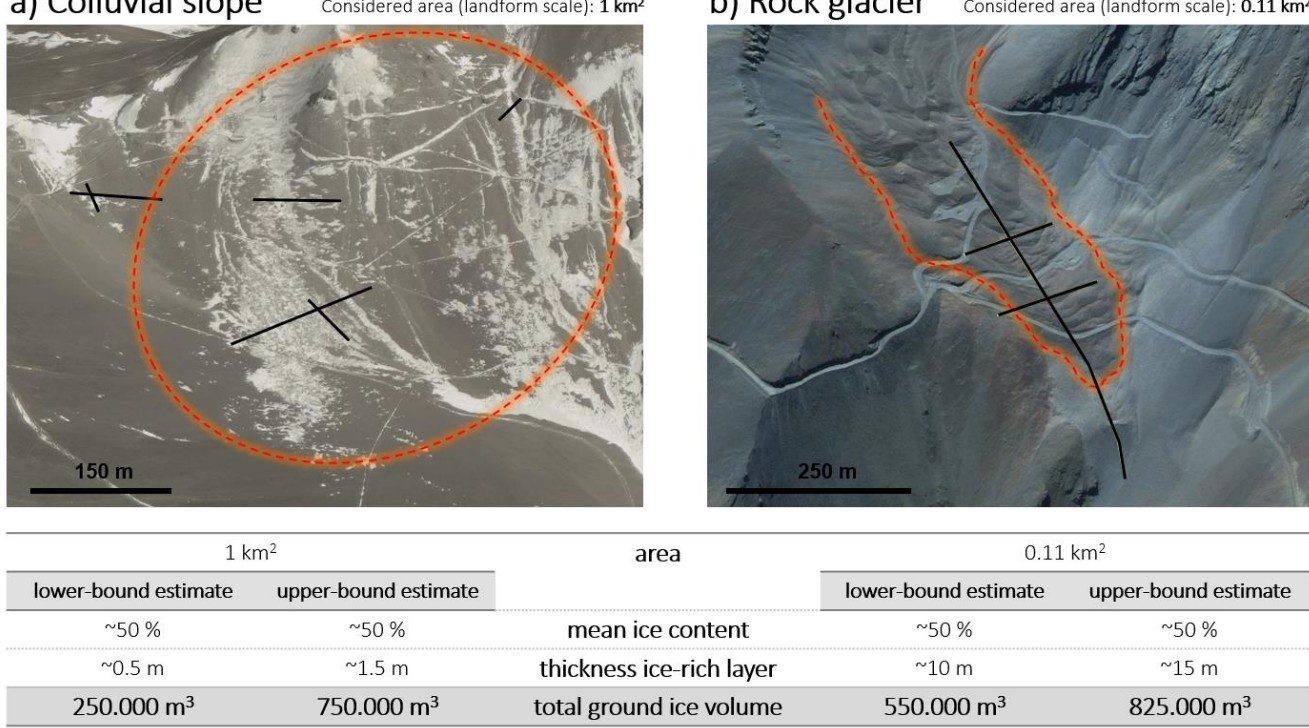

Figure 15: Estimated total ice content of a) a colluvial slope area containing profiles D03 - D08 versus b) a rock glacier area (profiles A01 - A03).

## 6    Conclusion and Outlook

Based on more than 50 geophysical surveys from various regions in the Central Andes, this study demonstrates the value of geophysical surveys to a) detect ice-rich permafrost occurrences in various landforms (also beyond prominent forms such as rock glaciers); and b) to estimate ground ice volumes in permafrost regions. The added value of combined ERT and RST surveys lies in an increased reliability of the interpretation (e.g., regarding the identification of bedrock), and the potential for ice content quantification through coupled petrophysical relationships such as within the 4-phase model.

The availability of various ground truthing data (cores from boreholes, test pits, natural outcrops) in this study allows the validation of the geophysical results for many cases. The good agreement between independent validation data and interpreted geophysical profiles confirms the detection of ice-rich layers in various non-rock-glacier permafrost landforms, emphasizing the value of geophysical data in the scientific debate on the role of ice-rich permafrost in the hydrological cycle. Further, we observe a substantial intra- and inter-site heterogeneity of the thickness of the ice-rich layer(s) and ice volumes, which is often wrongly inferred from visual inspections alone. Geophysics-based estimates on ground ice content therefore allow for more accurate assessments than purely remote-sensing-based approaches without a solid data basis. The data set presented in this paper is therefore one of the first available extensive set of field-based and validated data regarding the presence and total quantities of ground ice in the Central Andes.



The analysis of 52 ERT and 24 RST profiles within this study confirmed that ice-rich permafrost is not restricted to rock glaciers, but is also observed in non-rock-glacier permafrost slopes in the form of interstitial ice as well as layers with excess ice, resulting in substantial ice contents (e.g. D09, D04, C07), which can be close to the volumes observed in rock glaciers (D09). Consequently, non-rock glacier permafrost landforms, whose role for local hydrology has so far not been considered in remote-sensing based approaches, may, depending on the catchment size of the watershed, be similarly relevant in terms of ground ice content on a catchment scale and should not be ignored when quantifying the potential hydrological significance of permafrost.

On the other hand, a realistic estimate of ground ice volume is only the first step towards the evaluation of the hydrological importance of permafrost within a catchment. Further factors, such as a) different response times of permafrost landforms to observed and projected atmospheric changes in the Central Andes, and b) the dominance of the relevant hydrological processes (e.g. melting vs. sublimation, and discharge vs. evaporation), play a decisive role in the annual contribution to total runoff to downstream water resources from degrading permafrost (or to evaporation and sublimation) (e.g., Rivera et al., 2017). According to Duguay et al. (2015) the contribution of degrading permafrost to the total runoff of a catchment is difficult to measure, hence quantify, and therefore remains basically unknown. Without a reliable determination of these factors (e.g., by measuring or modelling the full energy balance over permafrost areas, cf. e.g., Harrington et al., 2018), the relevance of permafrost for the hydrological cycle remains strongly speculative. Preliminary modelling approaches suggest that this is negligible and would be non-measurable in the arid Andes (Arenson et al., 2013), and a recent analysis of mass-balance rates of ice masses in the Argentinian Central Andes confirms that rock glaciers showed almost zero mass balance rates from 2000 – 2018 (Ferri et al., 2020). However, no publications exist so far, that have specifically calculated the contribution of rock glaciers to streamflow in the semiarid Andes of Chile (Schaffer et al., 2019). Studies from other mountain environments (e.g., the European Alps, Marmy et al., 2016; Scherler et al., 2013) have shown that, depending on the snow cover and surface characteristics, the degradation of rock glaciers can be a very slow process because of the extremely efficient insulating effect of the active layer (coarse blocks) and the latent heat effect. Haeberli (1985) approximated the time needed for the complete decay of ice-rich permafrost in rock glaciers under a warming climate to be in the order of centuries to millennia, and Krainer et al. (2015) showed that ~10.000 years old permafrost ice persisted until today even during warm periods of the Holocene. The quantitative contribution of melting ground ice of degrading permafrost in rock glaciers to the annual discharge from the catchment can therefore be very small (Harrington et al., 2018; Krainer et al., 2015; Pruessner et al., 2021) and the relative contributions from other ice-poor permafrost landforms without blocky surfaces and thin but widespread ground ice layers still remain unknown. The geophysical data set presented in this study may therefore serve as input for modelling studies on the overall amount of ground ice present within the periglacial belt and estimates regarding the relative contributions of rock glacier and non-rock glacier ground ice to runoff in the semi-arid regions of the Central Andes.



## Acknowledgements

The acquisition of this comprehensive data set would not have been possible without the valuable support and hard work of
numerous field helpers from Chile, Argentina and Switzerland. Therefore, we sincerely thank all field helpers for their efforts
in the field. The authors also would like to acknowledge the support from various private companies that agreed for having
their data published, provided additional information, and logistically supported the various field campaigns.

## Code/Data availability

The data that support the findings of this study are available from the corresponding author upon request and will be provided
through an online data repository after acceptance of the manuscript.

## Author contribution

CHi planned, coordinated and participated at the geophysical campaigns, processed the geophysical data, conducted the 4-
phase modelling, wrote the major part of the text, and made all figures. CHa had the overall lead of the geophysical campaigns,
and contributed to the study design. CM coordinated and participated at two of the geophysical field campaigns, and helped
with data processing. PW and LA coordinated the environmental impact assessment studies, which included the geophysical
campaigns, but also borehole drilling, excavation of test pits and collection of other data. They planned and coordinated the
field logistics of the geophysical campaigns together with CHi, and provided further background information. All authors
contributed actively to the discussion and interpretation of all data sets, and the intermediate and final version of the manuscript.

## Competing interests

The authors declare that they have no conflict of interest.

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



**Appendix:**

Table A1: Overview about the data quality of the geophysical data. Abbreviations: n = number of quadrupoles, n_filtered = remaining quadrupoles after filtering, RMS = root mean square error.

| Profile | ERT | | | RST |
|---|---|---|---|---|
| | n | n_filtered (%) | RMS [%] | RMS [ms] |
| A01 | 360 | 92 | 7.7 | 2.9 |
| A02 | 1188 | 81 | 9.6 | 2.6 |
| A03 | 360 | 89 | 6.0 | 2.5 |
| A04 | 360 | 85 | 7.9 | |
| A05 | 636 | 100 | 1.8 | 2.7 |
| A06 | 636 | 91 | 3.6 | 2.6 |
| A07 | 360 | 99 | 7.0 | 1.7 |
| A08 | 912 | 96 | 3.0 | 1.7 |
| A09 | 360 | 99 | 2.6 | 2.0 |
| A15 | 360 | 100 | 9.0 | |
| A16a | 360 | 100 | 2.4 | 1.9 |
| A16b | 360 | 100 | 9.8 | 1.8 |
| A17 | 912 | 100 | 4.0 | |
| A24 | 636 | 95 | 2.1 | |
| A25 | 360 | 100 | 2.3 | 1.4 |
| B01 | 636 | 100 | 4.0 | 2.3 |
| B02 | 636 | 100 | 2.5 | 2.5 |
| B03 | 912 | 100 | 5.8 | |
| C02 | 636 | 100 | 4.3 | 1.2 |
| | | | 4.3 | 1.0 |
| C03 | 360 | 100 | 6.7 | |
| C04 | 360 | 100 | 1.3 | |
| C06 | 360 | 100 | 4.1 | |
| C07 | 912 | 100 | 3.0 | 1.2 |
| C08 | 360 | 100 | 3.2 | |
| C09 | 360 | 100 | 1.2 | |
| C10 | 360 | 100 | 1.5 | |
| D01 | 360 | 100 | 1.3 | |
| D02 | 636 | 100 | 1.1 | |
| D03 | 360 | 100 | 2.5 | |
| D04 | 360 | 100 | 6.1 | 1.3 |
| D05 | 912 | 100 | 2.3 | |
| D06 | 636 | 100 | 2.3 | |
| D06b | 360 | 99 | 2.0 | |
| D07 | 1740 | 99 | 2.4 | |
| D08 | 912 | 99 | 4.6 | |
| D09 | 360 | 99 | 2.8 | 1.1 |
| F10 | 360 | 99 | 2.9 | 1.6 |
| E03_A | 360 | 81 | 5.9 | |
| E03_BV | 636 | 100 | 2.7 | |
| E03_BH | 360 | 100 | 2.7 | |
| E04 | 636 | 99 | 3.8 | |
| E05 | 360 | 100 | 2.2 | |
| E08 | 912 | 99 | 5.0 | 2.0 |
| E09 | 360 | 94 | 6.6 | |
| E11_H | 360 | 100 | 2.5 | 2.1 |
| E11_D | 360 | 100 | 2.8 | |
| E12 | 360 | 100 | 3.4 | |
| E13 | 360 | 100 | 2.2 | |
| E14 | 360 | 97 | 1.6 | |
| E15 | 360 | 100 | 2.1 | |
| E16 | 636 | 97 | 4.8 | |
| E17 | 360 | 72 | 22.1 | |




**Figure A1: inverted tomograms of all ERT profiles from coarse-blocky sites (same spatial and colour scales), sorted by landforms**

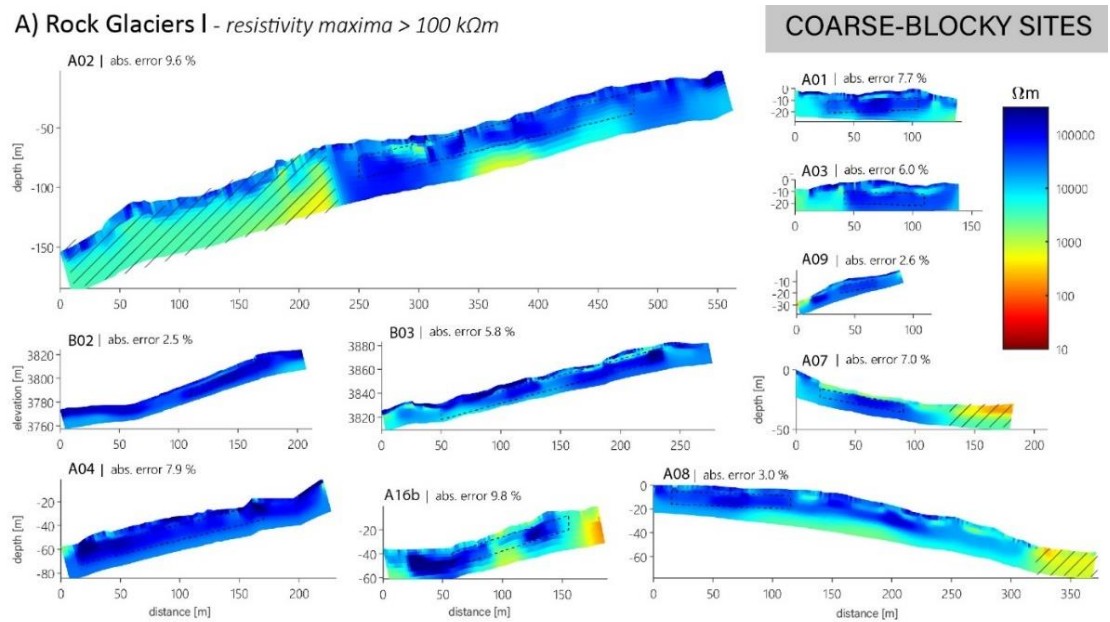

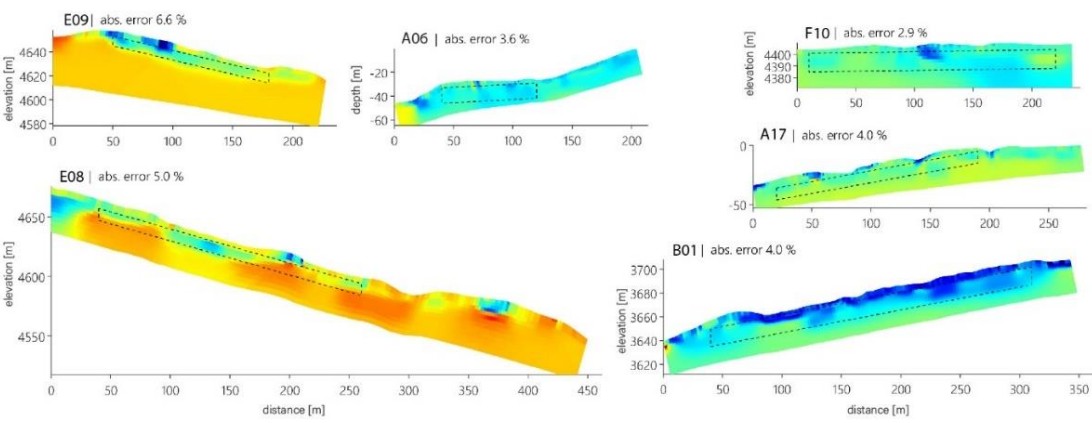

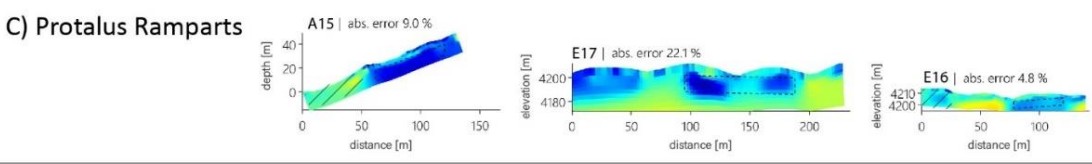

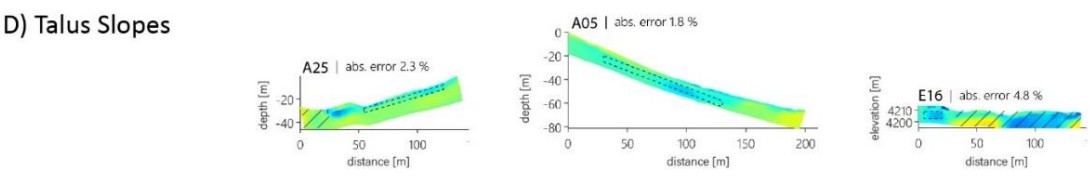





**Figure A2: inverted tomograms of all ERT profiles from sediment and bedrock sites (same spatial and colour scales), sorted by landforms**





**A3: inverted tomograms of all RST profiles from coarse-blocky sites (same scales and colour scales), sorted by landforms**

## A) Rock Glaciers

## B) Protalus Ramparts

## C) Talus Slopes



**A4: inverted tomograms of all RST profiles from sediment and bedrock sites (same scales and colour scales), sorted by landforms**

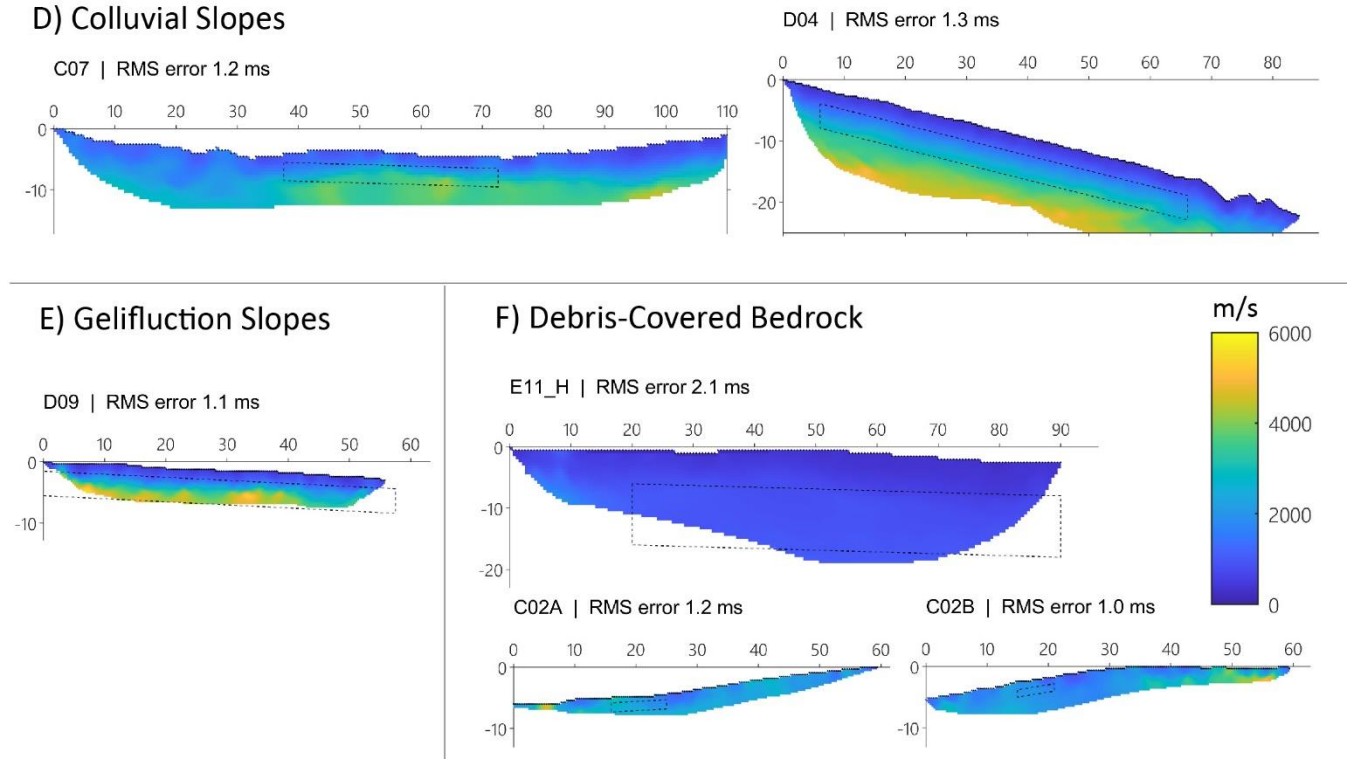