# Peer review of "Towards accurate quantification of ice content in permafrost of the Central Andes, part I: geophysics-based estimates from three different regions"

_The Cryosphere, 2021_

## Author Comment (AC1)

**Referee 2 (anonymous)**

*This paper is a summary of an extensive geophysical survey field effort at the Central Andes of Chile and Argentine where the objective was to study ground ice content. The authors analyze an extensive set of geophysical surveys (53 geoelectrical and 24 refraction seismic profiles) ranging from high latitude locations ranging between 28 to 32 °S. The dataset covers different landforms including ice-poor bedrock to ice-rich rock glaciers and the authors analyze the thickness of ice layers at each survey location. In my opinion, the paper is suitable for publication in this journal. Here are some specific comments that could improve the quality of the manuscript.*

**REPLY:** Thank you very much for your careful reading and the suggestions below, which we will address by point-by-point response in the following.

**C1:** *Line 40: Maybe replace "key stores" with "main storage"*

    **REPLY:** agreed and changed accordingly

**C2:** *Line 98: A closing parenthesis is missing.*

    **REPLY:** changed accordingly

**C3:** *Line 138: Two "therefore", consider changing wording of one if them.*

    **REPLY:** we rephrased the sentence accordingly

**C4:** *Line 364: ZOI abbreviation has already been explained. I suggest just using "ZOI". Same for PR and TS (Line 373).*

    **REPLY:** changed accordingly

**C5:** *Line 368: space after "10a,"? Same comment for Line 377.*

    **REPLY:** changed accordingly

**C6:** *Line 446-447. Is this sentence needed here? It seems misplaced and it is its own paragraph?! Consider "weaving in" somewhere else.*

    **REPLY:** We agree and combined it with the preceding paragraph as follows:
    *Therefore, we used ground truth data from the various field sites to validate the geophysical data. Table 3 gives an overview over the different types of available ground truthing data (boreholes, test pits, natural outcrops), the respective depth range covered, and the type of validation provided by the different data.*

**C7:** *Line 524: consider writing "the periglacial belt (see sections 4.2 and 4.3)" to replace "Examples are given in sections 4.2 and 4.3").*

    **REPLY:** changed accordingly

**C8:** *Table 1: CL and AR are not defined anywhere.*

    **REPLY:** We added the explanation of the abbreviation in the caption of the table (CL = Chile; AR = Argentina).

**C9:** *Lines 563-565. D09 are mentioned twice and it seems like D09 is for non rock and rock glaciers (can it be both?). First it refers to D09 as "non-rock glaciers" and the second time mentioned it says "observed in rock glaciers (D09)". Maybe there is a mistake in numbering?*

    **REPLY:** Thank you very much for spotting this. Yes, we agree that this is a bit misleading. We wanted to express, that the ground ice content of profile D09 is close to that of rock glaciers, but we will rephrase the sentence for clarity in the revised version.

***C10:*** *Figure 1: The scale bars are really hard to see. I suggest that these are made in white or boxed in.*

**REPLY:** agreed and changed accordingly

***C11:*** *Figure 2: Consider making lines thicker and/or in a different color (white?). They are really hard to see now. Also, the name of the lines are almost invisible. Is it possible to add an approximate scale bar? a, b, c and so forth are not mentioned in caption. Some of the text is really hard to read (i.e., text in b, d, g).*

**REPLY:** Thank you for this comment. We will increase the visibility of the lines in our revised version. However, as most profiles stretch over foreground and background parts of the photographs, a scale bar could not cover the different dimensions in a consistent way, and would be confusing in many cases. We therefore prefer not to do that. The length of the ERT profiles is indicated in Table 2, which may help the get a feeling for the dimension.

***C12:*** *Figure 3: Should it be a) rather than A)? Figure 2 uses a) but most figures use A).*

**REPLY:** We will homogenise the type of letters for subplots in the figures.

***C13:*** *Figure 5: It is mentioned in a "section of the ERT profile" but it isn't mentioned which section and this is true for all figures like this one. For example, I would have like to see, "section of the ERT profile covered by the RST profile (see x = A m to x = B m in Figure 3a doe full profile). Also, is the Ohmm scale different between Figure 5 and Figure 3? It seems like this is true for most of these comparisons.*

**REPLY:** We will indicate the position of the RST profile within the ERT profile for all these plots. We will also consider homogenising the colour scales.

***C14:*** *Figure 8: Same comment as for Figure 5. It is unclear what section (distance) of the profile shown in Figure 7.*

**REPLY:** see reply to C13.

***C15:*** *Figure 10: No c) in figure to the right.*

**REPLY:** will be added in the revised manuscript.

***C16:*** *Figure 13: and here "a" and "b" are written "(A)" and "(B)". Consider staying consistent between figures.*

**REPLY:** see reply to C12

***C17:*** *Figure 15: Consider labeling each line in the figure.*

**REPLY:** yes, we can do that.

***C18:*** *Appendix: Table 1: Consider writing "Data quality overview of the" rather than "Overview about the data quality of the ".*

**REPLY:** agreed and changed accordingly

***C19:*** *Figures A1 – A4: None of these captions start with a capital letter and there are no periods at the end of the captions.*

**REPLY:** agreed and changed accordingly

***C20:*** *Figure A3 (Line 800): Add "Figure" before "A3". Should A4 be A3? Also, add "Figure before "A4" (or A3).*

**REPLY:** agreed and changed accordingly

---

## Author Comment (AC2)

**Referee 1 (Jigjidsurengiin Batbaatar)**

*Dr. Hilbich and their colleagues measured resistivity and seismic wave propagation in large number of sites in the Central Andes. Their aim was to distinguish ice from their survey data. They also drilled through some of the landforms to check the validity of their geophysical surveys. It is a very data-rich paper and I commend their efforts. The authors state that they collected their data as part of "environmental impact assessment" commissions, and I am truly happy to see that the authors got the permission to publish these privately-funded surveys. It is rare to see such data published and made available to the public. I would like this paper to be published after the following general issues are addressed and the manuscript revised accordingly:*

**GENERAL COMMENTS (GC):**

**GC1:** *The main results of this paper, the ERT and RST profiles, appear to be published already in a series of reports (e.g., see my comments below for Ln 187–188). However, this fact was not explicitly stated/disclosed in the paper. The formatting of the references to the reports is incomplete and makes it hard to figure out the publisher of the reports and where to find the access to these reports. It is OK to summarize published results in addition to new results, but it would not be OK to present published data as new result. It is up to the editor how to handle this. Please consult with the editor. Please also make it clear in the text whether the data reported in this paper are new and never published before or not. It could be just my misunderstanding, but I get this notion from reading the text, so it's important to change the writing about this issue to avoid ambiguity.*

> **REPLY:** We acknowledge that it was not clearly stated how the data presented in this paper have been used prior to this publication. We therefore want to emphasize that these surveys have been conducted as part of multiple environmental impact assessments (EIA) for mining projects that are currently in planning in the Andes. These exploration projects are at different stages and typically such project developments take decades. The results of the surveys were summarized in internal reports that were prepared for those various project developers, but those data have not yet been made public. Some projects are currently under regulatory review and information can be requested by the public through those governmental agencies. We want to emphasize that the joint assessments presented in this paper are outside the scope of any regular EIA and are therefore considered unique. Because the environmental evaluation processes of the projects for which those surveys were completed are still ongoing it is not possible to provide more details in the manuscript. As indicated (Section Code/Data availability), the data that support the findings of this study can be made available upon request. In our revised manuscript we will emphasise more clearly that neither data nor their interpretation have been previously published, and we leave it to the editor to decide whether the internal (unpublished) reports should be mentioned or not). We will further add the following sentence: "The survey data and additional information can further be requested through the Servicio de Evaluación Ambiental in Chile or the Government of San Juan in Argentina". We also believe that the specific project doesn't matter to the overall message of the manuscript, and it eliminates some of the understandable emotions that are associated with the developments of large projects in the Andes.
> We hope that this response is acceptable and understandable.

**GC2:** *The interpretation of the tomographies and the results derived it depend on the 4-phase model (4PM) but the model was not described with enough detail. In other words, the very reason to accept the reported results as feasible is not provided properly. It may be familiar to geophysicists, but other readers don't know how 4PM distinguishes between the four "phases": rock, water, air, ice.*

> **REPLY:** Thank you very much for this comment. The 4PM was introduced and published in Hauck et al. (2011) and several publications with applications on permafrost, but we agree that the 4PM was not introduced sufficiently in the present publication. and will therefore add a specific paragraph in section 2 and the corresponding equations of the 4PM model in a specific appendix.

**GC3:** *Maximum and/or minimum values of the color scales used in the ERT and RST don't match the actual colors in the tomographies. Color blind readers, for example, may not perceive the differences between the colors*

*in the tomographies, and may use the values listed on the color scales to get the wrong impression that such values are expressed in the tomographies where it's not. Make sure every color scale actually matches the values of the tomographies.*

> **REPLY:** the colour scales were chosen to cover the overall range of all tomograms used in this study, and we used by intention the same resistivity/velocity range for all tomograms to facilitate intercomparing of different profiles. We consider this an important prerequisite for a reliable comparison of the entire data set. We understand, however, the concern of the reviewer. As a compromise, we will add the respective min/max resistivity or velocity values to the tomograms or indicate them in the colour bars.

**SPECIFIC COMMENTS (SC)**

**SC1:** *Ln 128–135: Please mention "ice, water, air and rock" being the four "phases" in the model at the first instance/sentence where you introduce it. Most readers with relatively less experience with geophysical survey methods (like myself) the four phases denote something else: solid, liquid, gas, and plasma. The confusion arises especially when I read "...consistent estimates of all 4 phases..." in Ln 134. Another point in this section is that it's not clear from text how PJI model achieves similar estimates of ice, water, air, and rock content without prescribing porosity. You didn't end up using this model in your research, but you still need to explain what PJI is to your non-geophysicist readers.*

> **REPLY:** Thank you for this remark. We realise now that our description of the ground ice quantification was too short in the original manuscript and that the 4PM (Hauck et al. 2011) and PJI (Wagner et al. 2019) models have to be better explained (see also our reply to the previous comment). We will add an additional section describing the models in more detail.

**SC2:** *Ln 142–143: Did you follow their approach to estimate the uncertainties in your data? If true, say so here, because there's no other place where you discuss the uncertainty calculations.*

> **REPLY:** Our approach is similar but not identical to the one presented in Halla et al. 2021. In our case we used a combination of two approaches to reduce the uncertainty in the comparative analysis of the ice contents for the different profiles: a) the ice content was only considered for the zones of interest (ZOI) defined for each profile, in order to pick representative zones of the tomogram within the permafrost layer (and to avoid distortions of the modelled ground ice content by including erroneous/ambiguous parts of the tomograms, or parts outside of the landform); and b) we always used three different 4PM runs spanning over the most probable porosity range for the respective landforms (SED: 30-45-60 %; TS: 40-50-60 %; RG: 40-60-80 %), and the resulting minimum and maximum ice content values were used as uncertainty range in the comparative analysis of all data (see Fig. 11b). By this, we believe that we can exclude important sources of error and narrow the uncertainty range.
> We agree that this has not sufficiently been explained in the Methods part of the paper, and we will add the corresponding explanations in the revised version.

**SC3:** *Ln 149: It is not clear how the ZOIs were selected for each profile. What does "assumed to be representative"? Who is making this assumption? You, the authors, or Etzelmuller et al. (2020)? This is a good example where you should use first-person narrative: "We selected ZOIs based on our assumption of this and that factors, following the previous approach by Etzelmuller et al." This is important because the characterization of icerich/ poor units in each profile relies on the selection of ZOIs, and you've spent only three sentences to explain what ZOIs are.*

> **REPLY:** This is an important point, which will be clarified in the revised version. The ZOI's were defined by the authors according to the study focus, i.e. the landform specific ice content. We moved one paragraph explaining our choice of ZOI's from the Results section to the Methods section, rephrased the paragraph accordingly and added some more details regarding the choice of ZOI's

**SC4:** *Ln 157–159: I commend the authors' effort to convince the private companies to release the data and make it available for the broader scientific community. I wish more private data would become public like this.*

> **REPLY:** Thank you for this comment. Negotiations with these companies have not been easy and therefore we are grateful that we are able to present in this paper what we have. We hope you understand the challenges and please see also our reply to GC1.

**SC5:** *Ln 187–188: Wait a minute--does "All profiles have been analyzed and interpreted" means the results of this paper was already published somewhere? Please clarify. The references cited after this statement all appear to be reports published in the city of Fribourg. Who is the publisher of these reports? How can the public access these reports?*

> **REPLY:** see our reply to GC1

**SC6:** *Ln 362–364: The actual description of ZOI is finally provided here in the Results section, not in the Methods section.*

> **REPLY:** we moved this description to the Methods section and enlarged it (see answer to SC3)

**SC7:** *Section 5.1: Ground truthing vs. tomographies*
*I appreciate the fact that the authors and earnestly discussing the shortcomings of the borehole and geophysical surveys. Would it be a big ask if I suggest you to provide your opinion on how incorporate these 'ground' data with remotely-sensed data? I'm only asking this because you've mentioned in the Introduction section that the remote sensing studies made their interpretations without consulting any 'ground' data.*

> **REPLY:** Thank you for this comment. Here, we can directly refer the reviewer to our companion paper (= part II of this publication):
>
>> Mathys, T., Hilbich, C., Arenson, L. U., Wainstein, P. A. and Hauck, C.: Towards accurate quantification of ice content in permafrost of theCentral Andes - Part II: an upscaling strategy of geophysical measurements to the catchment scale at two study sites (in review in TCD).
>
> In this accompanying manuscript we propose a methodology for the upscaling of the geophysical-based ground ice quantification to an entire catchment to estimate the total ground ice volume in the study areas. We show that (i) conventional ground ice estimates for rock glaciers without in-situ data may 3significantly overestimate ground ice contents, and (ii) substantial volumes of ground ice may also be present in catchments where rock glaciers are lacking.
> Another example of the combination of ground-based geophysical and e.g. kinematic data from InSAR time series is presented in Kunz et al. 2022 (https://doi.org/10.1016/j.geomorph.2021.108052). In the revised manuscript, we will indicate the potential of such ground-based data for remote-sensing approaches more prominently.

**SC8:** *Ln 533–534: The volume of ice in the colluvial slope depends on how you define its perimeter? Please explain why the extent of the colluvial slope is defined as it's shown in Figure 15. It looks rather an arbitrary ellipse. At least the satellite image in the background is so flat that the topography is not apparent.*

> **REPLY:** This figure represents an exemplary comparison of the overall dimension of ground ice content in different landforms. The perimeter indicated in Fig 15 represents an area of 1 km$^2$ within a well-studied colluvial slope (with a much larger total area), and this 1 km$^2$ area is only used to illustrate the spatial dimension of 1 km$^2$ colluvial slope in the landscape. For a more detailed analysis we refer to the companion paper Mathys et al. (Part II of this study). We will rephrase this part in the revised manuscript in order to be clearer.

**SC9:** *Ln 545–546: Show the North arrows in the pictures. Provide the source of the satellite images in the background. Indicate that these are not perspective images, but flat mapview of the satellite images.*

> **REPLY:** agreed and changed accordingly

**SC10:** *Ln 559: I commend all the efforts you've put into these geophysical surveys and drilling, and I agree that ground data is always better than the low-resolution remotely-sensed data. However, dissing the remote sensing studies as "approaches without a solid data basis" is not a helpful contribution to science. The remote sensing-based studies are firstorder estimates where there's no ground data exist, but they're based on reasonable assumptions about the ground conditions and their manifestation on the satellite data. Please educate us how remote sensing approaches should be improved using your data, instead of dismissing them with no consideration.*

> **REPLY:** We apologise to the reviewer if we have used a too impolite phrasing – we did not intend to disqualify remote sensing studies as we completely agree of their value in large-scale studies and in the absence of ground truth data! However, the phrase mentioned by the reviewer was taken slightly out of context as we

are only saying that "By this, geophysics-based estimates on ground ice content allow **for more accurate** assessments than purely remote-sensing-based approaches without a solid data basis." We are not saying that surveys without ground truthing are worthless, we are only indicating that it is better to have ground truthing. We suggest therefore to modify the sentence as follows: "Purely remote-sensing-based approaches can provide valuable first order estimates in the absence of ground-based data. However, geophysics-based estimates on ground ice content have shown to allow for more accurate assessments. The data set presented in …"

**DETAILED COMMENTS (DC) TO THE TEXT**

*Below are somewhat nit-picky comments I made regarding the grammatical and technical corrections. The paper will greatly benefit if the authors address these issues and revise the text/figure/tables accordingly:*

*DC1: Ln 45: This sentence is continued from "On the one hand" in the previous paragraph. It's not a good idea to form a new paragraph by this sentence.*

> **REPLY:** agreed and changed accordingly

*DC2: Ln 46: replace methodology with "methods". Methodology is the study of methods.*

> **REPLY:** Being non-native English speakers, we are not sure which expression is more adequate in this context. However, "methodology" is a widely-used term in scientific papers, which is why we chose it here.

*DC3: Ln 45–48: Maybe place the references after each of the reasons for the disputes. In other words, take the references after "...is disputed" and put after the reasons to make it clear which reference is cited for what reasons of dispute.*

> **REPLY:** agreed and changed accordingly

*DC4: Ln 57–59: It's not fair to say "with only very few exceptions" when there are plenty of ground measurements, especially done by local scientists and reported in Spanish. It's not that the ground data doesn't exist, maybe it's that the existing data is not integrated enough into the newse measurements of ice in periglacial landforms. For example, Trombotto et al. (2020: https://doi.org/10.1002/ppp.2044) show that the amount of groundwater flowing into the rock glaciers may dictate the ice content in rock glaciers. I agree with more field observations needed to to better constrain the ice content. Please rephrase the sentences in Ln 53–59.*

> **REPLY:** agreed and changed accordingly

*DC5: Ln 92 and everywhere else: Please replace methodology with "methods". Methodology is a study of methods. For example, the last sentence in this paragraph would read like this: "This study of methods include..." where in fact you meant to say "These geophysical methods include..."*

> **REPLY:** see our comment to DC2.

*DC6: Ln 109 and elsewhere: Please use first-person forms as much as possible. Directly say "We collected ERT data in the field using..." It's unambigious who did what. Sometimes, especially Discussion section, such third-person narrative may cause confusion whether it's referring to data collected by the current authors or from a cited reference.*

> **REPLY:** agreed and changed accordingly

*DC7: Ln 124: Which spelling is correct for REFLEXW or Reflex-W (in Ln 108). Please make them consistent.*

> **REPLY:** agreed and changed accordingly

*DC8: Ln 125–127: Please be consistent in reporting the uncertainty/error. For ERT it's reported in % but it's reported in ms for RTS. Another point is that the RST is presented in velocity, m/s, but here's it's the return time I believe, in ms. Please that what the actual physical parameter/property it's referring to. Don't just say "... RMS...were below 3 ms for RST..." leaving the readers to guess what it's referring to. The RMS in the RST*

*reported in % in Figure 8, for example. I suggest reporting all errors in the unit they were presented in figures, ohm-m and m/s, and use it consistently throughout the text.*

**REPLY:** Thank you for spotting the inconsistency between RMS errors given in ms and % for the RST data. We will homogenise and give all RMS errors in %.

*DC9: Ln 186: The "profiles" cannot be measured. Maybe say "We measured electric resistivity along 19 profiles on..." Again, please use first-person narrative to be more specific about who did what.*

**REPLY:** agreed and changed accordingly

*DC10: Ln 186–187: I'm guessing the numbers in the parentheses are number of profiles for each landform from the fact that 16+3, but it's not clear at all. Please fix and don't leave anything to guess for the readers.*

**REPLY:** agreed and changed accordingly

*DC11: Ln 308: What is the source of this hydrothermal condition? Is there active volcano/vent nearby? What other evidence you have that the bedrock could have been hydrothermally altered?*

**REPLY:** In their overview paper, Davidson and Mpodozis (1991) show, for example, that there are epithermal gold deposits in the area that are of Cretaceous to Cenozoic age. The ores and the volcanic rocks with which they are associated were generated at a non-collisional, ocean-continent convergent plate margin. However, the complex tectonic history that played a part in controlling the development of the hydrothermal systems and major ore deposits along the Chile – Argentina border resulted from changes in the dynamics of the lithospheric plates, primarily throughout the Cenozoic.

Davidson, J. and Mpodozis, C. 1991. Regional geological setting of epithermal gold deposits, Chile. In: Econ. Geol.  v86, pp. 1174-1186.

*DC12: Why are all tables embedded into the manuscript as images? Turning the numerical and textual data into images makes it hard to recognize via text-to-speech programs for disabled readers, and significantly reduces the data discoverability, which may lead to lower citation otherwise. I don't know if the authors actually provided their data in editable tables, but the PDF conversion turned them images. Make sure that the data tables are provided as readable numbers and text in the published version.*

**REPLY:** agreed and changed accordingly

*DC13: Ln 389: As far as I can read from Figure 10C, there's no bedrock there. Talus, rampart, rock glaciers are all depositional landforms made of sediments. Clarify what you mean by "...higher P-wave velocities (bedrock) is apparent...".*

**REPLY:** Thank you for this comment. This sentence stems from a former version of this figure/paragraph and is now confusing. We will change the text accordingly.

*DC14: Table 1: Provide explanations for the other abbreviations in addition to the various landforms (ERT: Electrical Resistivity Tomography; RST: Refraction Seismic Tomography; CL: Chile; AR: Argentina)*

**REPLY:** agreed and changed accordingly

*DC15: Table 2: Provide unit for altitude (m asl?)*

**REPLY:** agreed and changed accordingly

*DC16: Table 3: Change to "Overview of" not 'over'. I'm guessing 'y' means yes and 'n' means no. Please change the text so that it's clear that 'y' is for confirmed, and 'n' is for not confirmed. Don't leave anything to guessing.*

**REPLY:** agreed and changed accordingly

*DC17: Figure 1:*
*- Enlarge the font size of the lat/long in the main index panel*
*- Include the units and direction of the lat/long values. In other words, replace negative numbers with no units with 70°W and 30°S etc.*

*-Indicate the north orientation in each of the panels A–F. If north is up on all maps, please say so in the figure caption.*

**REPLY:** agreed and changed accordingly

***DC18:** Figure 2:*
*- Very nice. Clearly shows the variety of the landscape. This is a nit-picky stuff, but I think it looks better and grammatically correct if the second words were not capitalized. Protalus rampart in panel D is hard to read, maybe move the label to the right over the sky?*

**REPLY:** agreed and changed accordingly

***DC19:** Figure 3:*
*- Include the name of the parameter (resistivity) next to the color scale.*
*- Please make the second words small-cap (i.e., Rock glaciers I; Protalus ramparts etc).*
*- I don't understand the purpose of the label "COARSE-BLOCKY SITES". If the intention was to indicate that these profiles are select few from only sites with large boulder covers, it's better to say so in the figure caption. Because of the placement of this label, it gives an impression that that the profiles A01, A03, A09 and A07 are coarse-blocky sites and the the rest are rock glaciers I.*

**REPLY:** agreed and changed accordingly

*- Is there a really good reason for why Y-axes are elevation in some profiles, and depth in others? It is visually wrong and makes hard for the readers to compare the profiles. Please make them all in depths. I understand it's important to indicate the elevation here, where landscape can change rapidly depending on elevation, so maybe indicate the elevation of the top surface in each profile?*
*- The values of depth in profile A15 is positive, where every other depth values are negative as it should be. Does this mean the profile A15 should be flipped? Please fix.*

**REPLY:** We will homogenise the Y-axes of all profiles, but we prefer elevations, as this is important in the context of the data, and as some of the plots are very small, there is not much space for additional text.

***DC20:** Figure 5:*
*- Include the name of the parameters (resistivity and velocity) next to the color scales (vertically) in the top two panels.*
*- Spell out ERT and RST, even if it's repeated many times in the text. If you think the labeling of the parameters next to color scale is enough, you may even not need to say ERT and RST.*
*- Nit-picky stuff: Don't capitalize "Ice content". It's not an abbreviation like in ERT and RST. Move the "Ice content" next to the color scale (vertically.)*

**REPLY:** agreed and changed accordingly

*- The minimum of the color scale for ohm-m in panel A goes down to very dark red corresponding to 10^2, but I don't see such dark red on the profile. Please confirm the min and max values shown in the profile and make sure that the color scale is extended between the actual values shown in the profile. Otherwise, the color scale might give a false impression that the ERT goes all the way down to 10^2, because this rainbow color scale can be hard to separate the colors especially for color-blind readers.*

**REPLY:** see our reply to GC3

*- There's plenty of white space--don't abbreviate "absolute" in panel A and "average" in panel C.*
*- The labels inside the panels appear small (e.g., much smaller than the numbers on the axes). Please consult the journal's guide on figures and make sure these labels are big enough when they're shrunk and fitted into the publicaiton format.*

**REPLY:** agreed and changed accordingly

***DC21:** Figure 6: Include a label on the figure indicating which of these three pieces is top and which is bottom. Otherwise, the photo looks like three different cores, not three pieces of a single core. Caption says the upper part is ice, but it leaves the reader to guess that the upper salmon-colored piece is the one with massive ice.*

*Most of the readers of The Cryosphere distinguish ice with crystalline texture from sediment with granular texture, but there are readers who can't tell. Don't leave anything to their guess. The transition between the ice and sediment-rich segment is really nice in the photo! The gradient of the green hue in the middle of the core can even be used to estimate the abundance of the sediment in the ice. Really nice!*

> **REPLY:** agreed and changed accordingly

*DC22: Figure 7:*
*- Include the name of the parameter (resistivity) next to the color scale.*
*- I don't see any intense dark red colors in the profiles. Adjust the color scale to correctly show the actual minimum and maximum values represented in the profiles.*

> **REPLY:** see our reply to GC3

*- Replace the elevation in panel for E16 to depth, the same as the other two profiles. Otherwise, explain in the caption why E16 is shown in elevation and the other two in depth.*
*- What is the purpose of including zones not related to a talus slope? Particularly in E16 almost 80% of the profiles is marked with diagonal lines. I am OK with the profile A25 where most of it actually shows the talus slope. I understand that you wanted to keep the expensive data acquired in the field and publish it, but you need to explain in the caption what kind of landform it represents and what is the (approximate) composition and texture (gravel? sand? till? fan?).*

> **REPLY:** Profile E16 stretches over a talus slope in the beginning and further over two protalus ramparts (the same profile but other parts of it is used in other parts of the study), so we prefer to keep the entire profile and just indicate the important part.

*DC23: Figure 8: Similar comments I've made before*
*- Add labels to the color scales (resistivity, velocity, ice content) and remove the in-panel labels (ERT, RST, ICE CONTENT.) Please treat the color scales if you would with axes of any other plots.*
*- Adjust the color scale in panel A (dark red for 10^2) to reflect the actual values in the profile. I don't see intense dark red in the profile. Wrong color scale can mislead colorblind readers.*
*- In panel B, I don't see intense yellow color in the profiles as indicated in the color scale. The max value in the color scale goes all the way to 5000 m/s, but in the RST no pixel reaches that value. Please adjust the color scale to reflect the actual values presented in the profile.*

> **REPLY:** see our reply to GC3

*- Don't capitalize "Ice content"*
*- don't abbreviate absolute (in panel A) and average (in C).*
*- the font size of the in-panel labels could be too small.*
*- there's a "leftover" small cross-lines near the 160 m mark in panel C. Fix it.*

> **REPLY:** agreed and changed accordingly

*DC24: Figure 9: Add labels to the color scales (resistivity, velocity, ice content) and remove the in-panel labels (ERT, RST, ICE CONTENT.)*

> **REPLY:** agreed and changed accordingly

*- Now it's the other way around--the dark blue in the color scale for the maximum value is not present in the actual ERT profile. It seems like you've plotted all the profiles using the same color scale, which is inherently correct thing to do and it works when you show different profiles in the same figure, such as in Figure 3. However, when you show the profiles one at a time like in this figure and borrow the universal color scale, it may give an impression that the maximum ERT reaches 10^5 which is not true and misleading. I'm NOT suggesting to re-plot the profiles or use a different color scale. Instead, I'm suggesting just to cut off the unnecessary part of the color scale to reflect the actual values in the profiles.*

> **REPLY:** See our reply to GC3: we prefer to keep the colour bars as originally presented for better cross-comparison but add the min/max values to each plot

***DC25:*** *Figure 10:*
*- Be consistent in the Y-axis value for the resistivity in panel A. I think it's better to express it as 10^2, 10^3 etc, instead of a bunch of zeros or 1e+06.*
*- A nit-picky suggestion for the large panel in the right: spell out the landforms instead of the abbreviations (e.g., Sediment slope (SED); Talus slope (TS) etc.) I see that it's explained in the caption and the abbreviations are consistent with the panels on the left. However, you have plenty of white space to fit those words and it is always best if a figure itself can be understood clearly on its own without relying heavily on captions. Don't force the readers to hunt for what these abbreviations stand for.*
*- The same rule applies to ZOI. I highly recommend spelling out the zone of interest(s) instead of its abbreviation. I totally understand that it's near the end of the paper and a reader who started from the beginning memorized it already. However, imagine a reader who's skimming and just looking at the figures without the captions, to see if the paper is interesting. Your paper will attract much more people if the individual figures were clear on its own without heavily relying on text or captions.*

> **REPLY:** Thank you for the suggestion. We changed the y-axis accordingly. However, regarding the legend and the ZOI we believe that in this case the explanation in the figure caption is sufficient and that too much text in the figure would not improve the readability of the figure significantly.

*- Spell out ER and RS in the caption to avoid repeating "tomogram". Change it to "...respective electric resistivity and refraction seismic tomograms..."*
*- There's no 'C' in the scatter plot. Add C in the corner of the scatter plot.*

> **REPLY:** agreed and changed accordingly

***DC26:*** *Figure 11: Spell out PF. You have plenty of space to fit permafrost. In panel B, I like the way you've used gray and black lines for the error bars to indicate which is for max and mean values. The error bars for D009, however, look like drawn with the same gray color.*

> **REPLY:** agreed and changed accordingly

***DC27:*** *Figure 12:*
*- Remove the title of the figure "Max/Min ice content..." from the top of the figure and include it in the caption.*
*- Capitalize i in "Ice" in Y-axis*
*- Spell out "borehole" and "4-phase model" instead of BH and 4PM. Shorten the black and gray bar in the legend and you'll have plenty of space to fit the words.*
*- Add "survey sites" or "survey lines" in the X-axis. Otherwise, they're not explained in the caption and the readers need to hunt for what they are in the text. Again, make the figure stand on its own without relying on text and caption.*

> **REPLY:** agreed and changed accordingly

***DC28:*** *Figure 13:*
*- Add "Profile" before A02 and A16b as in "Profile A02".*
*- Why numbers in A02 and A16b are smaller than the letters? Fix them to have the same font size.*
*- I highly recommend adding a legend/explanation panel for all the lines, just like you have it in Figures 10 and 11. It could say:*
*- Borehole data*
*- 4-phase model estimates:*
*40% porosity*
*60% porosity*
*80% porosity*
*Calibrated with borehole data*

> **REPLY:** agreed and changed accordingly

*- I'm a bit worried about the fact that borehole data is represented as continuous black line from top to bottom. I think it's better to show the ice content estimates from the core with dots (or squares, diamonds etc) in their reflective depths, to truthfully reflect the method. Say in the caption what was the sampling interval in the core*

*for measuring ice content. What was the depth interval for the 4PM model runs? Indicate that in the caption too.*

**REPLY:** We use a sonic drill, which provided continuous cores and as such we estimated the volume ice content continuously. The estimates shown in Figure 13 are from the in-situ assessments of the drill core. We acknowledge that the estimates are in-situ estimates and based on comparison with laboratory data, the estimate error is ~10 %, which was added in the figure caption. Nevertheless, we think that our representation of the data is not misleading. The 4PM model discretisation was added to the caption.

***DC29:*** *Figure 14:*
*- Please spell out PR and RG. You have plenty of space there.*
*- A tiny orange polygon is visible below the black bar in E17. Must be an error in illustrator.*
*- Change in the caption to "...the minimum (black) and maximum (grey) thickness of..."*
*- I highly recommend including a legend/explanation for this figure showing what black, gray and orange bars and the black lines indicate. If somebody looked at this figure without the caption, they will not be able to what it entails at all. That means other people cannot use this figure for presentations and other visual reports straight away, diminishing the possible exposure and citation.*

**REPLY:** agreed and changed accordingly

***DC30:*** *Appendix: All figures in the appendix lack caption. Provide figure captions even for the appendix figures. You need to explain the what the dashed-line boxes are etc.*

**REPLY:** agreed and changed accordingly

*- Add an appendix table that includes lat/long and elevation of all profiles. I would like to, for example, to load these locations into Google Earth and check them out. For future researchers as well, it would be easier to find and re-measure the active layer thickness, permafrost depth, and ice content etc. Your field data are the very first measurements, which is laying the ground for the repeat measurements in 100 or 200 years in the future.*

**REPLY:** See our answer to GC1. We hope that you understand that we cannot supply this table.

***DC31:*** *Table A1: Maybe report the RST error in m/s, not ms? As for the ERT, it would be more useful to see the errors in ohm-m than in relative percentages. You have room in this table—why not report the errors both in the physical units and percentages?*

**REPLY:** For ERT, an absolute error (ohm-m) would be quite meaningless, as it depends mainly on the overall resistivity range of the respective tomogram and would not represent a comparable measure. RMS error values in % are therefore the standard in ERT applications. Similarly, the RMS error of RST data describes the residual between observed and calculated travel times, and quantifying it in terms of m/s (P-wave velocity difference) would be meaningless. However, we will correct the unit for the RMS error for the RST data into % (see also reply to DC8).

***DC32:*** *Figure A1: it's the same figure as Figure 3. No need to include it in the appendix.*

**REPLY:** No, Figure A1 includes also talus slopes in addition to the three other landforms. Therefore, it is added to the appendix – as the counterpart to Figure A2 (sediment and bedrock sites).

***DC33:*** *Figure A2:*
*- Please provide the parameter name and unit next to the color scale (resistivity)*
*- The maximum values of the color scale with intense dark blue do not appear on the actual profiles. Adjust the color scale to reflect the actual values on the profiles.*

**REPLY:** see comments above and the reply to GC3

*- Profile E11_D has "depth" axis but the values indicate it's actually elevation.*
*- Add "depth [m]" in Profile A24 after fixing the Y-axis of E11_D*

**REPLY:** agreed and changed accordingly

***DC34:*** *Figures A3 and A4:*

*- Provide the name and unit (depth [m]?) of the Y-axes*
*- The horizontal axes also lack name and unit too (distance [m]?)*
*- The RMS errors are given in ms (millisecond) but I think they should be m/s. Please fix. If it's indeed ms for re-turn times, it's not really helping because the profiles are in velocities [m/s]. How do you think the reader should convert return times to velocities and get the correct sense of RMS error?*

> **REPLY:** see comments to DC8 and DC31. The RMS value (in %) is a standard measure for the reliability of the inversion result (tomogram showing velocities in m/s). It quantifies the difference between the observed and calculated travel times. Otherwise, the calculated and observed travel times would need to be plotted for each tomogram (similarly to the pseudo-sections of observed and calculated apparent resistivities for ERT data), which is far beyond the scope of such a paper. Researchers working with ERT and RST data understand how to read the respective RMS values.

---

## Author Response (AR2)

**Comments to the author**:
Dear Christin Hilbich and coauthors,

Thank you for your revised submission and for addressing the comments and suggestions of both reviewers. Below I make some additional suggestions to improve the clarity of your manuscript. I appreciate the efforts you have made to present this comprehensive dataset, which promises to make a lasting contribution to the field.

With best wishes, Huw Horgan

*Dear Editor,*

*Thank you very much for your suggestions to improve clarity and language of the paper. We strongly appreciate this, and we adjusted/corrected the text as proposed for most of your suggestions. They are visible in the document with the track-changes-mode. In the case of three of your comments, please find a separate reply below.*

*With our best wishes,*

*Christin Hilbich (on behalf of all co-authors)*

Minor editorial points. Some of these are suggestions, some are grammatical corrections.

L9. 'In view of...' Avoid sentence constructs like this wherever possible as they detract from the point you are trying to make. I suggest: 'Water scarcity in the Central Andes due to ongoing climate change means that the significance of permafrost occurrences for the hydrologic cycle is controversial.'

*Reply: we slightly modified your proposition, which now reads as: "Increasing water scarcity in the Central Andes due to ongoing climate change recently induced a controversial debate on the significance of permafrost occurrences for the hydrologic cycle."*

L24 'We state the geophysics...' I appreciate your point but encourage you to consider the tone of this sentence. I don't think it is your intent to devalue remote sensing studies and gloss over the limitation of field geophysics. Your text is much more nuanced than this sentence implies. I suggest something like 'We show that field geophysics based estimates of ground ice content, while more labor intensive, are considerably more accurate than remote sensing approaches.'

*Reply: thank you for this comment. It was not our intention to devalue remote sensing studies, so we are happy for your proposition and took it over like this.*

L163 '...the context of the projects has no further relevance for the scientific content of this paper' I appreciate what you wish to state here but suggest you include what the context is, even though you have already stated it earlier. > 'the Environmental Impact Assessment context of the projects has no further relevance for the scientific...'
*Reply: we corrected the text like this: "Apart from the fact that some of the considered permafrost landforms had surface disturbances (e.g., access roads or drilling platforms), the context of developing environmental impact assessments for mining projects has no further relevance for the scientific content of this paper."*

---

## Author Response (AR3)

**Comments to the author**:

Dear Christin Hilbich and Coauthors,

I am pleased to accept your manuscript for publication in The Cryosphere. Please make the (very) minor technical corrections prior to final submission. I will not need to review your contribution again.

Congratulations and best wishes,

Huw Horgan

Minor technical corrections:

L9. 'induced controversial debate' to 'induced controversy and debate'. Otherwise 'controversial' applies to the 'debate', meaning the way the debate was carried out was controversial. I think you mean there was controversy and debate.

L105 'method' to 'methods'

L118 'Overview over' to 'Overview of' or 'Summary of'

L396 'The striking pattern... is apparent' to 'A striking pattern is apparent, with...'

*Reply:*

*Dear Editor. We thank you for reading the manuscript carefully again and for correcting the text. We agree with all your suggestions and changed the text/wording accordingly.*

*Thanks a lot for your support and for accepting our manuscript for publication in TC!*

*Best regards,*

*Christin Hilbich and Co-authors*